# Dysregulated *SASS6* expression promotes increased ciliogenesis and cell invasion phenotypes

Eleanor Hargreaves[1] , Rebecca Collinson[2] , Andrew D Jenks[3] , Adina Staszewski[2,4], Athanasios Tsalikis[4,5] , Raquel Bodoque[5,6], Mar Arias-Garcia[5] , Yasmin Abdi[2], Abdulaziz Al-Malki[5,7], Yinyin Yuan[5,8] , Rachael Natrajan[9] , Syed Haider[9] , Thomas Iskratsch[10] , Won-Jing Wang[11] , Susana Godinho[12] , Nicolaos J Palaskas[13] , Fernando Calvo[14] , Igor Vivanco[15] , Tobias Zech[1] , Barbara E Tanos[2]

Centriole and/or cilium defects are characteristic of cancer cells and have been linked to cancer cell invasion. However, the mechanistic bases of this regulation remain incompletely understood. Spindle assembly abnormal protein 6 homolog (SAS-6) is essential for centriole biogenesis and cilium formation. SAS-6 levels decrease at the end of mitosis and G1, resulting from APC[Cdh1]-targeted degradation. To examine the biological consequences of unrestrained SAS-6 expression, we used a non-degradable SAS-6 mutant (SAS-6ND). This led to an increase in ciliation and cell invasion and caused an up-regulation of the YAP/TAZ pathway. SAS-6ND expression resulted in cell morphology changes, nuclear deformation, and YAP translocation to the nucleus, resulting in increased TEAD-dependent transcription. SAS-6–mediated invasion was prevented by YAP down-regulation or by blocking ciliogenesis. Similarly, down-regulation of SAS-6 in DMS273, a highly invasive and highly ciliated lung cancer cell line that overexpresses SAS-6, completely blocked cell invasion and depleted YAP protein levels. Thus, our data provide evidence for a defined role of SAS-6 in cell invasion through the activation of the YAP/TAZ pathway.

## Introduction

Centrosome and cilium abnormalities have been shown to be involved in cancer development (Han et al, 2009; Wong et al, 2009; Godinho et al, 2014; Godinho & Pellman, 2014; Jenks et al, 2018) and cancer cell invasion (Godinho et al, 2014). Accordingly, the primary cilium–dependent hedgehog pathway is an important regulator of malignant progression and invasion (Jing et al, 2023).

Spindle assembly abnormal protein 6 homolog (*SASS6* gene, SAS-6 protein) is a key factor in centriole assembly and duplication (Leidel et al, 2005), a process that requires a sequence of coordinated events involving a group of six proteins (Delattre et al, 2006; Pelletier et al, 2006; Sugioka et al, 2017). SAS-6 together with SAS-5/STIL forms the core of the cartwheel structure, a geometric scaffold that defines procentriole radial symmetry (Nakazawa et al, 2007; Kitagawa et al, 2011; van Breugel et al, 2011).

Although the function of SAS-6 in centriolar biology has been well studied, how SAS-6 might regulate various cancer-associated phenotypes remains relatively unexplored. Work by Shinmura et al showed that *SASS6* overexpression was associated with mitotic chromosomal abnormalities and poor prognosis in patients with colorectal cancer (Shinmura et al, 2015). *SASS6* is overexpressed in a number of human cancers, including kidney cancer, bladder cancer, and breast invasive carcinoma (Shinmura et al, 2015), and reportedly promotes proliferation by inhibiting the p53 signaling pathway in esophageal squamous carcinoma cells (Xu et al, 2020). Interestingly, knockdown of *SASS6* reduced the growth of MDA-MB-231 triple-negative breast cancer cells (Du et al, 2021). STIL, a known interactor of SAS-6, has been shown to localize to the cell cortex and promote changes in the actin cytoskeleton that support invasion (Liu et al, 2020). In addition, Polo-like kinase 4 (PLK4), a kinase that promotes SAS-6 recruitment to the cartwheel structure (Dzhindzhev et al, 2014; Ohta et al, 2014; Moyer et al, 2015; Arquint &

[1]Institute of Systems, Molecular, and Integrative Biology, University of Liverpool, Liverpool, UK   [2]Centre for Genome Engineering and Maintenance, Department of Biosciences, College of Health, Medicine and Life Sciences, Brunel University London, London, UK   [3]Division of Molecular Pathology, The Institute of Cancer Research, London, UK   [4]Department of Surgery and Cancer, Faculty of Medicine, Imperial College London, London, UK   [5]Chester Beatty Laboratories, Institute of Cancer Research, London, UK   [6]Translational Research Unit, Hospital General Universitario de Ciudad Real, Ciudad Real, Spain   [7]Department of Pathology, George Washington School of Medicine and Health Sciences, Washington, DC, USA   [8]Department of Translational Molecular Pathology, Division of Pathology and Laboratory Medicine, The University of Texas MD Anderson Cancer Center, Houston, TX, USA   [9]The Breast Cancer Now Toby Robins Research Centre, The Institute of Cancer Research, London, UK   [10]School of Engineering and Materials Science, Faculty of Science and Engineering, Queen Mary University of London, Engineering Building, London, UK   [11]Institute of Biochemistry and Molecular Biology, College of Life Science, National Yang-Ming University, Taipei, Taiwan   [12]Centre for Cancer Cell and Molecular Biology, Barts Cancer Institute, Queen Mary University of London, London, UK   [13]David Geffen School of Medicine, University of California, Los Angeles, CA, USA   [14]Instituto de Biomedicina y Biotecnología de Cantabria (Consejo Superior de Investigaciones Científicas, Universidad de Cantabria), Santander, Spain   [15]Institute of Pharmaceutical Science, King's College London, London, UK

Correspondence: barbara.tanos@brunel.ac.uk

 Life Science Alliance

Nigg, 2016), is overexpressed in cancer (Godinho et al, 2014; Godinho & Pellman, 2014) leading to centrosome amplification, increased cell invasion (Godinho et al, 2014), and increased small vesicle secretion (Adams et al, 2021).

Studies of the function of SAS-6 in cancer are complicated by its periodic degradation at the end of mitosis/G1 by the APC^Cdh1 complex via its KEN box (Strnad et al, 2007). To prevent the cyclic loss of SAS-6 expression, we transduced cells with a nondegradable SAS-6 KEN box mutant (SAS-6ND) that was expressed throughout the cell cycle (Fong et al, 2014).

We found that SAS-6 overexpression led to an increase in cilium numbers and cilium length. Interestingly, analysis of The Cancer Genome Atlas (TCGA) showed that high levels of *SASS6* expression were consistent with poor prognosis in adrenocortical carcinoma, low-grade glioma, and kidney, liver, and lung cancer patients. Given that metastatic disease is generally associated with poorer prognosis (Ganesh & Massague, 2021), this raised the question of whether increased SAS-6 levels could be associated with metastasis. Indeed, SAS-6 overexpression showed increased invasion that could be suppressed by blocking ciliogenesis. Our transcriptome analysis revealed that the expression of SAS-6ND resulted in the activation of YAP/TAZ, a pathway associated with metastasis (Piccolo et al, 2023). Notably, blocking YAP/TAZ function reverted SAS-6–induced invasion. Analysis of cell morphology in SAS-6–overexpressing cells showed cell flattening and nuclear deformation, features previously shown to open the nuclear pore complex and promote YAP nuclear import (Elosegui-Artola et al, 2017). A lung cancer cell line (DMS273) with endogenous SAS-6 overexpression was found to be both highly invasive and highly ciliated (80%). In this cell line, RNAi-mediated knockdown of SAS-6 significantly reduced YAP levels and completely abolished its invasion phenotype. Thus, our work shows a unique and novel function of SAS-6 in invasion through the regulation of YAP/TAZ and provides rationale for interrogating the therapeutic potential of targeting SAS-6 as a strategy to prevent metastatic disease.

# Results

## SAS-6 promotes an increase in ciliogenesis

SAS-6 is a key protein in centriole duplication. In cycling cells, SAS-6 protein levels oscillate throughout the cell cycle, decreasing in G1 upon degradation by the APC^Cdh1 complex that recognizes a KEN box (Strnad et al, 2007). Removing the conserved KEN box domain (Fig 1A) results in SAS-6 stabilization and overexpression (Fong et al, 2014). To maintain consistent levels of SAS-6 protein throughout the cell cycle, we transduced cells with a SAS-6 mutant (SAS-6ND) with the KEN box replaced with alanines (AAA), and placed it under the regulation of a tetracycline-inducible promoter (Fig 1A and B) (Fong et al, 2014). As expected, endogenous SAS-6 was not detected at centrioles in G1 in RPE-1 cells (Fig 1C), whereas SAS-6ND was expressed throughout the cell cycle in doxycycline-treated cells (Fig 1D). In ciliated cells, SAS-6ND localized to the proximal end of both mother and daughter centrioles (Fig 1E).

Given the localization of SAS-6 to mother centrioles, which can function as basal bodies, we examined whether SAS-6 could affect cilium formation. Interestingly, SAS-6ND expression resulted in increased cilium length and cilium number in RPE-1 cells (Figs 1F and G and S1A). Additional experiments in human mammary epithelial cells (HMECs) and Ras-transformed MCF10AT1 showed a similar increase in ciliogenesis (Fig S1B and C).

In our experimental system, the overexpression of WT SAS-6 also promoted an increase in ciliogenesis, likely because of saturation of the degradation machinery (Fig S1B and C). Doxycycline induction was maintained for 6 d and did not result in centrosome amplification (Fig S2A–C) in either SAS-6ND or parental RPE-1 cells (Fig S2A–C). Doxycycline treatment for 3 or 6 d in RPE-1 parental cells did not result in increased ciliogenesis compared with cells transduced with SAS-6ND (Fig S3A–C), which showed significantly more cilia that appeared longer (Fig S3A–C). Thus, the presence of SAS-6 supported an increase in cilium formation.

## *SASS6* expression is associated with poor prognosis in adrenocortical carcinoma, low-grade glioma, and kidney, liver, and lung cancer

Centriole and cilium proteins have been shown to play a role in cancer (Bettencourt-Dias et al, 2011; Godinho & Pellman, 2014). Particularly, *SASS6* overexpression was found to correlate with poor prognosis in a number of tumor types (Shinmura et al, 2015; Xu et al, 2020), highlighting the relevance of SAS-6 in cancer progression. Interestingly, knockdown of *SASS6* reduced proliferation of the highly aggressive/invasive MDA-MB-231 cell line (Du et al, 2021). Given the potential role of SAS-6 in cancer (Shinmura et al, 2015; Du et al, 2021), we examined TCGA database in search of clinical correlates. High *SASS6* expression resulted in lower survival probability over a 10-yr period. Our analysis revealed that the increased expression of *SASS6* correlated with poor prognosis in adrenocortical carcinoma, low-grade glioma, renal papillary cell carcinoma, and hepatocellular carcinoma and confirmed previous reports of poor prognosis in kidney renal clear cell carcinoma and lung adenocarcinoma (Shinmura et al, 2015) (Fig S4A–E) (Table S1). Metastatic disease is the primary cause of cancer death, and poor prognosis is often associated with increased invasion and metastasis (Ridley, 2011). We therefore wanted to investigate whether SAS-6 overexpression could be linked to invasive potential in cancer cells.

## SAS-6 overexpression leads to invasion that depends on the presence of cilia

The ability of cancer cells to migrate is a key process in metastasis (Yamaguchi & Condeelis, 2007) and a hallmark of cancer (Hanahan & Weinberg, 2011). Metastasis requires cancer cells to leave the tissue of origin through invasion and migration, enter blood vessels, and colonize distant tissues (Yamaguchi & Condeelis, 2007). Cell migration is a multistep process initiated by cell protrusions (Ridley, 2011). Formation of protrusive structures is driven by coordinated actin polymerization at the leading edge of the cell (Yamaguchi & Condeelis, 2007). We carried out a cell protrusion assay using microporous filters (Mardakheh et al, 2015). Briefly,

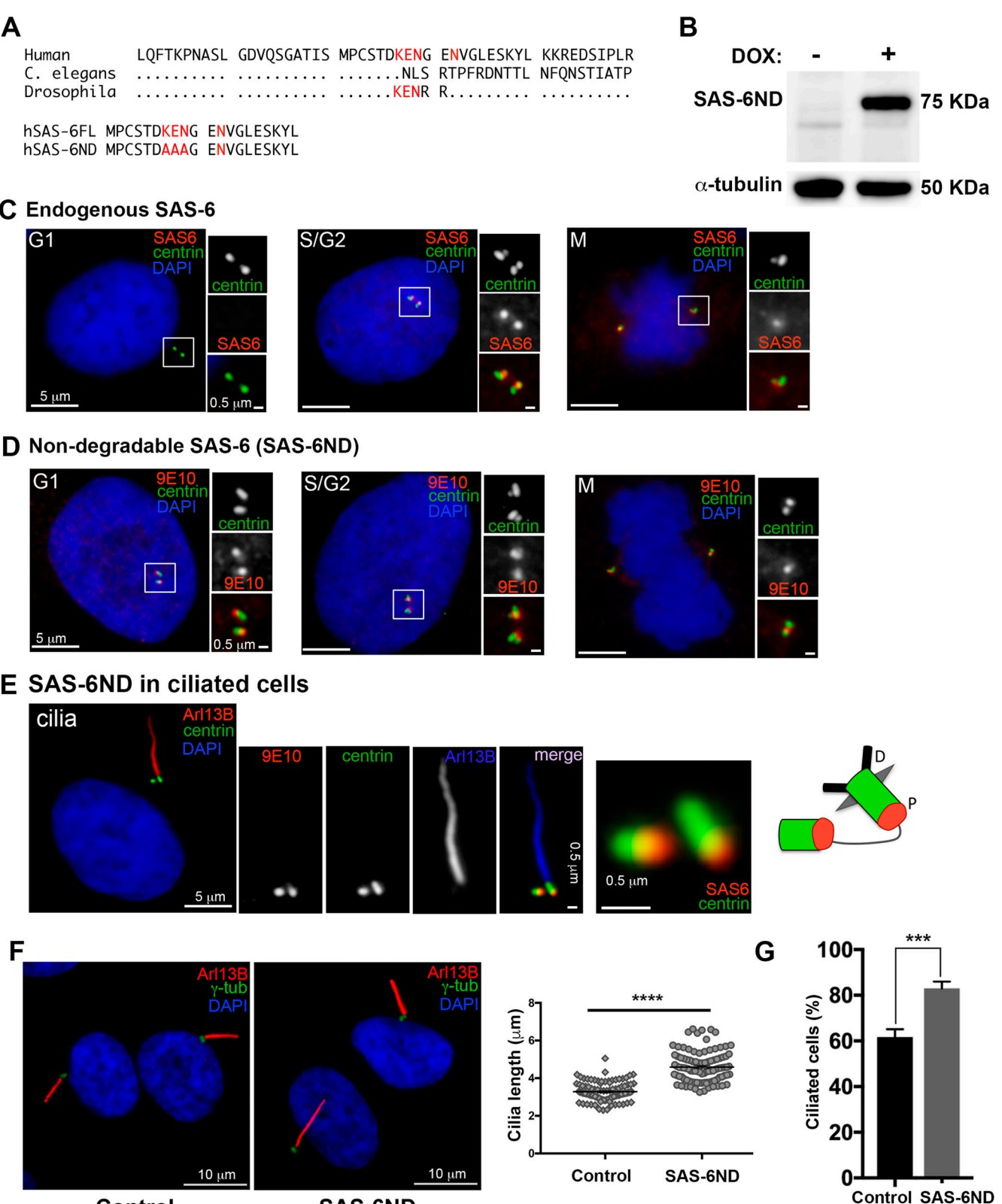

**Figure 1. Nondegradable SAS-6 (SAS-6ND) is stable throughout the cell cycle and promotes increased ciliogenesis.**
**(A)** SAS-6 protein structure showing a stretch of human SAS-6 protein and the conserved KEN box (shown in red). hSAS-6ND was generated by replacing the KEN box with an alanine stretch (red). **(B)** Western blot showing the expression of SAS-6ND (top) with alpha tubulin as a loading control. **(C)** Endogenous SAS-6 expression at different cell cycle stages (indicated). RPE-1 cells expressing GFP-centrin (green), stained with an antibody for endogenous SAS-6 (red). DNA is marked with DAPI (blue).

cells were seeded on top of collagen-I–coated 3-µm polycarbonate transwell filters in serum-free medium. After cells were attached, medium in the bottom chamber was replaced with a 10% serum counterpart to induce cell protrusions for 4 h. Using three different clones of cells with inducible SAS-6ND, we found that doxycycline treatment induced an increase in protrusions (Fig 2A). We then carried out a three-dimensional collagen invasion assay (Gadea et al, 2008; Sanz-Moreno et al, 2008). Cells were plated in a mixture of collagen in serum-free medium, spun down to the bottom of a 96-well filter, and placed at 37°C to allow collagen to polymerize. After this, medium with serum was added to the top of the wells and cells were allowed to move toward serum for 24 h before fixing them with 4% PFA. A higher number of SAS-6ND cells invaded through collagen toward serum compared with the uninduced controls (Fig 2B). This increase in invasion was also observed in Ras-transformed MCF10AT1 cells overexpressing either SAS-6WT or SAS-6ND (Fig 2C). Centriole and cilium signaling pathways play a role in cell migration (Rosario et al, 2015; Kazazian et al, 2017; Jing et al, 2023). We asked whether the invasion phenotype driven by SAS-6ND required the presence of primary cilia. To test this, we depleted SCLT1, a protein that we previously showed to be required for ciliogenesis (Tanos et al, 2013). As expected, removal of SCLT1 (Fig S6A) resulted in a decrease in primary cilia (Fig S6E–G), without affecting the cell cycle or resulting in centrosome abnormalities (Figs S5 and S6B–D). Our results showed that depletion of cilia via SCLT1 removal completely suppressed the invasion phenotype observed upon the expression of SAS-6ND (Fig 2D), suggesting that the presence of cilia is necessary for this process.

## SAS-6 invasion phenotype is associated with the activation of the YAP/TAZ pathway

To understand what molecular programs were activated by SAS-6 to promote invasion, we carried out a transcriptomic analysis. cDNA was obtained from untreated cells and doxycycline-induced cells expressing SAS-6ND and used as probes for DNA microarray hybridization. Gene set enrichment analysis (GSEA) revealed significant enrichment of genes involved in YAP/TAZ pathway activation after the overexpression of SAS-6 (Fig 3A and B). YAP is a transcriptional coactivator promoting transcription downstream of the TEAD promoter. The YAP/TAZ pathway has been shown to be activated in cancer and to promote invasion and cell proliferation (Piccolo et al, 2023). We found that among the genes most up-regulated in the YAP/TAZ gene set were CTGF and CYR61 (Fig 3C). Both of these genes encode matricellular proteins up-regulated downstream of YAP (Piccolo et al, 2023). We validated these results using qRT-PCR. Consistently, CTGF and CYR61 mRNA levels increased 20-fold after SAS-6ND induction (Fig 3D and E).

Cytoplasmic YAP is inactive and eventually phosphorylated by LATS and targeted for degradation, whereas active YAP localizes to the nucleus and activates transcription (Piccolo et al, 2023). We estimated the extent of YAP accumulation in the nucleus by quantifying the nuclear/cytoplasmic ratio of YAP using fluorescence microscopy. SAS-6–overexpressing cells showed a marked increase in nuclear YAP (Figs 3F, S7A–C, and S8C and D). Consistently, a luciferase-based reporter assay for TEAD-dependent transcription showed an increase in reporter activity upon the expression of SAS-6ND (Figs 3H and S7C and D). Parental RPE-1 cells, on the other hand, had a slightly higher baseline for nuclear YAP, which was unaffected by doxycycline treatment (Fig S8A and B). Thus, we reasoned that uninduced versus induced SAS-6ND–transduced cells represented a more physiologically relevant system given their isogenic background. Because SAS-6 seemed to promote the activation of the YAP/TAZ pathway, and considering the reported role of this pathway in cell invasion (Piccolo et al, 2023), we reasoned that blocking the YAP pathway would revert the invasion phenotype. To test this, we carried out a collagen invasion assay in the presence or absence of verteporfin, a YAP inhibitor that disrupts the YAP-TEAD interaction, decreases YAP expression, and blocks transcriptional activation of downstream targets of YAP (Liu-Chittenden et al, 2012; Brodowska et al, 2014). Treatment with verteporfin decreased YAP levels in cells overexpressing SAS-6ND (Fig 3I) and completely reverted the invasion phenotype (Fig 3J), suggesting that SAS-6–mediated invasion is dependent on YAP. siRNA treatment to specifically down-regulate YAP in SAS-6ND–expressing cells showed that depleting YAP completely blocked SAS-6–dependent invasion (Fig S9A–C). This also resulted in a net increase in YAP S127 phosphorylation relative to total YAP levels (Fig S9C–G). Because this particular LATS-driven phosphorylation takes place in the cytoplasm (Piccolo et al, 2023), this suggests that the remaining YAP protein is mostly cytoplasmic.

## Depleting SAS-6 in ciliated metastatic lung cancer cells blocks invasion

To find clinically relevant models, we mined the DepMap database for patient-derived cell lines with high SAS-6 expression. DMS273, a metastatic lung cancer cell line, showed comparably higher-than-average SAS-6 protein levels despite normal levels of mRNA (Fig 4A). A comparison of protein expression showed that DMS273 had over three times the level of endogenous SAS-6 compared with parental RPE-1 cells and uninduced RPE-1 SAS-6ND cells (Fig 4B and C). Staining for Arl13B, a bona fide ciliary marker, showed that DMS273 was highly ciliated, even in the presence of serum (Fig 4D and E). In DMS273, the cilium was only traceable with Arl13B and not with acetylated tubulin. We tested the invasive properties of these cells and found they were highly invasive (Fig 4F and G). Reducing SAS-6 protein levels using shRNA (Fig 4H) significantly reduced invasion to almost negligible levels (Fig 4F

Note that endogenous SAS-6 is absent in G1. **(D)** Nondegradable SAS-6 (SAS-6ND) expression throughout the cell cycle. Centrin-GFP is shown in green, and Myc-tagged SAS-6ND (9E10-antibody) is shown in red. DAPI is shown in blue. **(E)** SAS-6ND expression in ciliated cells. The primary cilium is marked with Arl13B (red in the main panel, blue in the inset). Centrin-GFP is shown in green, and Myc-tagged SAS-6ND is shown in red. DAPI is shown in blue. An inset with both the mother centriole (basal body) and the daughter centriole is shown, with a cartoon depicting SAS-6 localization (right). D stands for *distal* and P for *proximal*. **(F)** Ciliation in control cells and cells expressing SAS-6ND. A quantification of cilium length is shown in the right panel. *t* test, *P* < 0.0001. Note the increase in cilium length in the presence of SAS-6ND. **(G)** Quantification of cilia number in control cells and cells expressing SAS-6ND. *t* test, *P* < 0.001. Data are representative of five independent experiments.

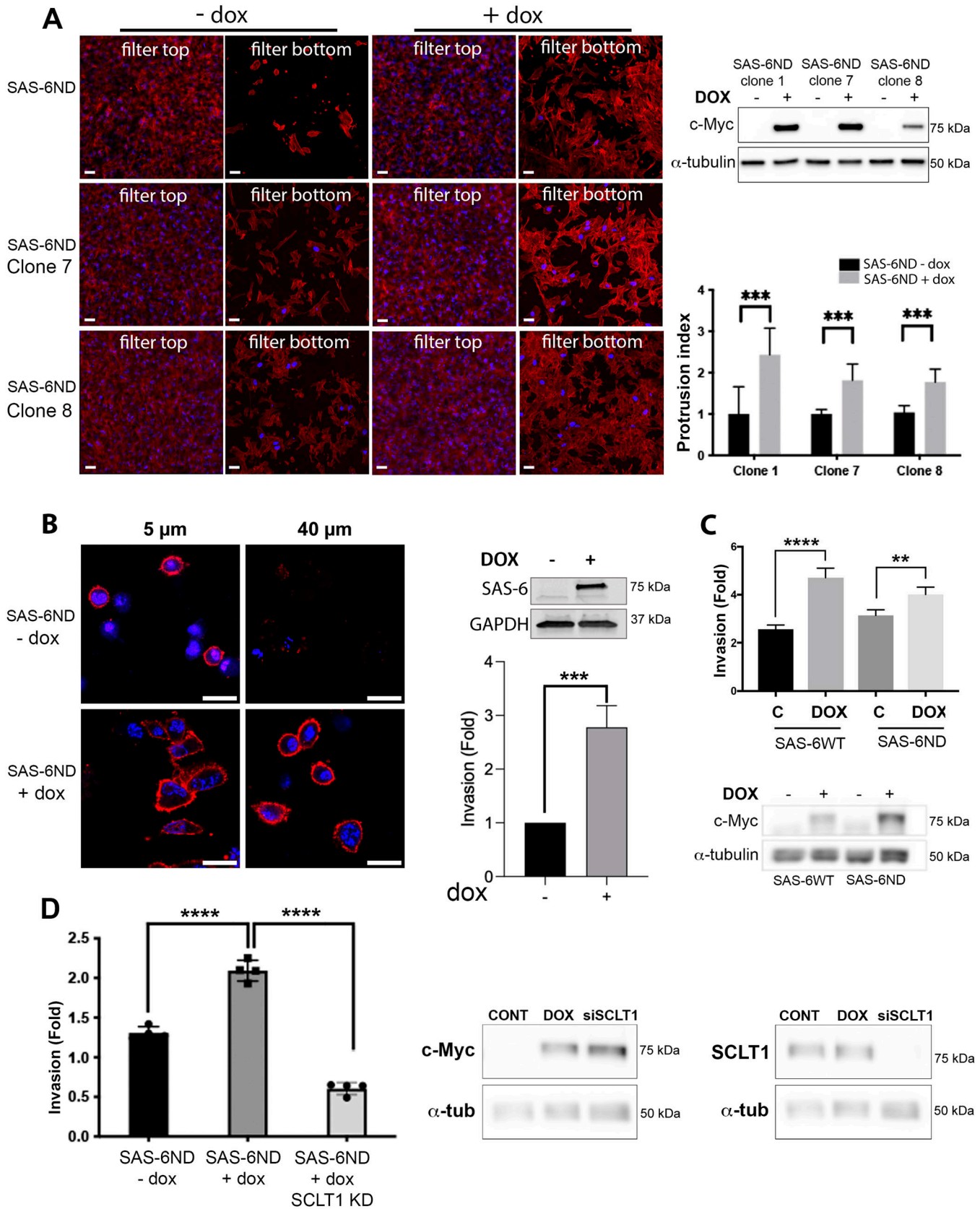

and G). SAS-6 down-regulation also significantly reduced total YAP protein levels and resulted in an increased ratio of YAP P-S127 to total YAP (Fig 4I–K). These findings support a role of SAS-6 in cell invasion through activation of the YAP/TAZ pathway.

### SAS-6 induces reversible changes in cell morphology

Multiple solid cancers have shown an up-regulation of YAP, which correlates with increased invasion, malignancy, and relapse (Piccolo et al, 2023). We examined the morphology of the actin cytoskeleton after induction of SAS-6 overexpression. SAS-6ND–expressing cells appeared flattened (Fig 5D) and showed increased actin alignment and condensed stress fibers (Fig 5A–C), consistent with an invasive phenotype. Analysis of the vertical and horizontal axes in the nucleus revealed a decreased nuclear aspect ratio, which confirmed their flattened appearance (Fig 5D and E). Further analysis showed decreased nuclear solidity and nuclear form factor and increased nuclear compactness (Fig 5F–H). Down-regulation of YAP in SAS-6–overexpressing cells reverted these changes in nuclear morphology (Fig S9H and I).

## Discussion

Our results collectively show that SAS-6 overexpression leads to increased ciliogenesis, YAP translocation to the nucleus, changes in cell morphology, increased TEAD-dependent transcription, and activation of the YAP/TAZ pathway. This results in an invasive phenotype both in nontransformed (and also RAS-transformed) cellular models and in patient-derived cancer cells (Fig 5I). Blocking ciliogenesis, SAS-6 expression, or the YAP/TAZ pathway reverts this phenotype.

SAS-6 has been extensively characterized as a key regulator of centriole structure, supporting the initial steps of the establishment of the ninefold symmetry and the recruitment of microtubules during centriole assembly (Leidel et al, 2005; Delattre et al, 2006; Pelletier et al, 2006; Nakazawa et al, 2007; Kitagawa et al, 2011; van Breugel et al, 2011). Here, we find that SAS-6 overexpression drives increased ciliogenesis in a number of cell lines. A positive role of SAS-6 in cilium formation is supported by work in *Caenorhabditis elegans* showing that a microcephaly-associated *SASS6* mutation results in shorter phasmid cilia (Bergwell et al, 2023). In *C. elegans*, centrioles initiate cilium assembly but degenerate at later stages of ciliogenesis (Serwas et al, 2017).

Unviable *sas-6* mutants transduced with a GFP-SAS-6 transgene regain embryonic viability and cilium formation (Serwas et al, 2017), as seen by a dye-fill assay (Hedgecock et al, 1985).

Consistent with previous research (Shinmura et al, 2015; Xu et al, 2020), we found that *SASS6* expression was associated with poor prognosis in a number of tumors, including lung cancer (Fig S4). Metastatic disease is the primary cause of cancer death, and poor prognosis is often associated with increased invasion and metastasis (Ridley, 2011). Notably, we found that cells overexpressing SAS-6 had increased cell protrusions and were more invasive (Fig 2). To test whether cilia affected this invasion phenotype, we depleted SCLT1 (Tanos et al, 2013) (Fig S6) and found that this prevented SAS-6–induced invasion (Fig 2). How mammalian SAS-6 might contribute to cell invasion has never been addressed. SAS-6-like (SAS6L), a paralog of SAS-6, localizes to the apical complex in Apicomplexa phylum, including Toxoplasma, Trypanosoma, and Plasmodium (de Leon et al, 2013; Wall et al, 2016). This complex is assumed to play a mechanical and secretory role during invasion in Apicomplexa to support host penetration and invasion (Gui et al, 2023). STIL, a protein that functions together with SAS-6 in centriole duplication (Nakazawa et al, 2007; Kitagawa et al, 2011; van Breugel et al, 2011; Arquint & Nigg, 2016), is present at the leading edge of the cell and affects cell motility and the actin cytoskeleton (Liu et al, 2020).

The overexpression of Plk4, the kinase upstream of SAS-6, has been shown to promote centrosome amplification, chromosomal instability, and cancer cell invasion (Godinho et al, 2014; Godinho & Pellman, 2014). However, we did not observe centrosome amplification in our experimental system (Figs S2 and S6) compared with the overt rosettes and de novo–formed centrioles previously seen with amplified Plk4 (Kleylein-Sohn et al, 2007). This suggests that the observed invasion phenotype cannot be the result of centrosome amplification.

To understand what drives the SAS-6 invasion phenotype, we analyzed changes in gene expression. GSEA showed a significant enrichment of YAP/TAZ pathway–related genes (Fig 3), which were accompanied by a significant translocation of YAP to the nucleus (Figs 2F and G, S7, and S8). We confirmed pathway activation with a qRT-PCR for CTGF and CYR61, two YAP-regulated genes, and with a reporter assay for TEAD-dependent transcription (Figs 3H and S7D). Treatment with the YAP inhibitor verteporfin, or siRNA-mediated YAP down-regulation reverted SAS-6–driven invasion (Figs 3I and J and S9).

To confirm these observations in a disease-relevant model, we identified a metastatic lung cancer cell line, DMS273, with high

---

**Figure 2. SAS-6 overexpression leads to invasion that depends on the presence of cilia.**
**(A)** Cell protrusion assay in RPE-1 cells transduced with tetracycline-inducible SAS-6ND. Confocal image of cell protrusions in 3-$\mu$m transwell filters for three different SAS-6ND–overexpressing clones (indicated). Panels show representative images of the top and the bottom of the filter. A control, non–doxycycline-treated condition is shown. The actin cytoskeleton is marked with phalloidin (red). DAPI marks DNA (blue). Scale bar, 10 $\mu$m. A Western blot with the levels of Myc-tagged SAS-6ND expression is shown on the top right. $\alpha$-Tubulin is used as a loading control. The graph shows the quantification of cell protrusions in the presence of SAS-6ND. Error bars represent the SD. *t* test is indicated, ∗∗∗*P* < 0.001. **(B)** Collagen invasion assay in SAS-6ND–overexpressing RPE-1 cells. Panels show confocal images at 5 and 40 micron. Actin is shown in red, and DNA is marked with Hoechst in blue. Scale bar, 20 $\mu$m. A Western blot with SAS-6ND levels is shown on the top right, and GAPDH is used as a loading control. A quantification of invasion is shown in the lower right. *t* test, ∗∗∗*P* < 0.001. Data are representative of four independent experiments. **(C)** Collagen invasion assay in MCF10AT1 cells, overexpressing either SAS-6WT or SAS-6ND (indicated). A Western blot with the expression of the Myc-tagged construct is shown (lower panels) with a c-Myc Western blot and $\alpha$-tubulin loading control (indicated). *t* test, ∗∗*P* < 0.01, ∗∗∗∗*P* < 0.0001. Data are representative of two independent experiments with four replicas. **(B, D)** Collagen invasion assay in RPE-1 cells as shown in (B), upon down-regulation of SCLT1, a protein required to form cilia. The invasion index is shown on the right. Error bars represent the SD. *t* test is indicated, ∗∗∗∗*P* < 0.001. Data are representative of two independent experiments with four replicas. Western blots show the levels of Myc-tagged SAS-6ND and SCLT1 levels (indicated). $\alpha$-Tubulin is used as a loading control.

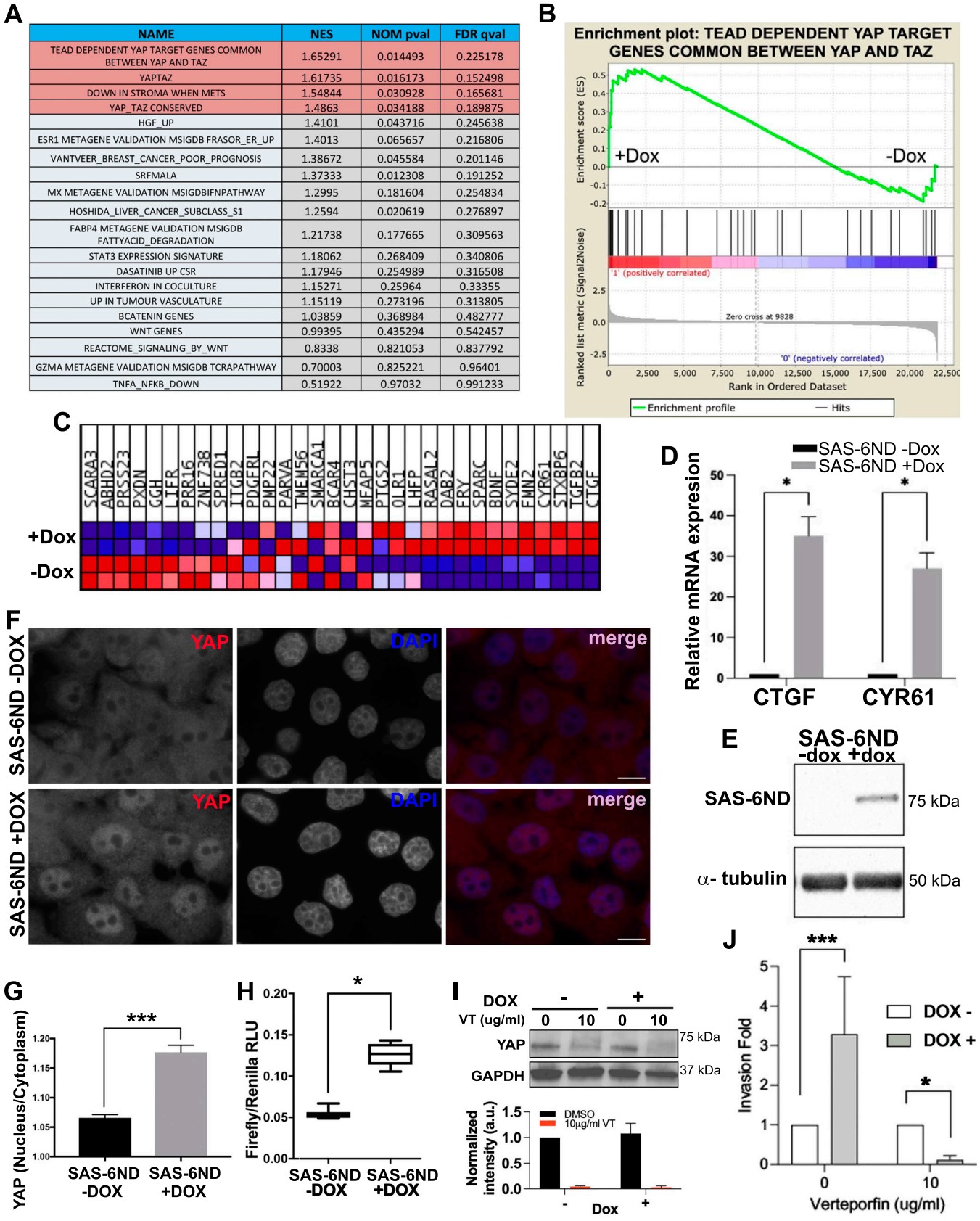

SAS-6 expression (Fig 4A–C). This cell line was highly ciliated, even in the presence of 10% serum (Fig 4D and E). shRNA-mediated SAS-6 down-regulation in these cells significantly blunted their invasiveness (Fig 4F–H). We interrogated the fate of the YAP pathway in these cells and found that SAS-6 down-regulation significantly reduced the levels of transcriptionally active YAP (Fig 4I–K).

The YAP/TAZ pathway has been shown to be activated by mechanical forces (Piccolo et al, 2023) wherein cell flattening is sufficient to promote the opening of the nuclear pore complex leading to the nuclear accumulation of YAP (Elosegui-Artola et al, 2017). We observed actin cytoskeleton changes and cell flattening in SAS-6–overexpressing cells (Fig 5). On the other hand, YAP down-regulation successfully reverted nuclear compression in SAS-6–overexpressing cells (Fig S9H and I). Further experiments will show whether SAS-6 supports the cytoskeleton changes needed to activate YAP or whether nuclear YAP promotes these changes upon SAS-6 overexpression.

It is possible that signals generated by SAS-6 could support actin cytoskeleton changes that drive cell invasion. Work by G. Gupta uncovered an interaction between SAS-6 and FAM21 (Gupta et al, 2015), a component of the WASH complex, which assembles at centrioles (Visweshwaran et al, 2018) (reviewed in Fokin & Gautreau [2021]) and has a direct role in Arp2/3 activation, cell protrusion, actin remodeling, and invasion (Zech et al, 2011). Furthermore, Ofd1, a SAS-6–interacting protein (Gupta et al, 2015; Go et al, 2021), was recently shown to regulate a centriole/cilium-dependent signal for Arp2/3 complex activation, suggesting that centrioles could coordinate invasion cues (Cao et al, 2023). Therefore, one possibility is that SAS-6 cooperates with or activates the WASH complex to regulate invasion.

Our results are consistent with a model for the regulation of invasion by SAS-6 that involves morphological changes, nuclear deformation, and YAP translocation to the nucleus. These changes likely drive the activation of TEAD-dependent transcription to support cell invasion (Fig 5I).

# Materials and Methods

## Cell lines and reagents

All cell lines were maintained at 37°C in a humidified atmosphere containing 5% $CO_2$. Human telomerase–immortalized retinal pigment epithelial cells hTERT-RPE-1 or RPE-1 were purchased from the American Type Culture Collection (ATCC) and transduced with pBABE retro GFP-centrin 2. RPE-1 and DMS273 cells were cultured in Dulbecco's modified Eagle's medium/Nutrient Mixture F-12 Ham (DMEM/F-12; Sigma-Aldrich) media containing 0.365 mg/ml L-glutamine, 15 mM Hepes, 1.2 mg/ml sodium bicarbonate (NaHCO3) and supplemented with 10% FBS (Gibco) and 1% penicillin/streptomycin (Gibco). 1 µg/ml doxycycline was added to growth media for 6 d before experiments to induce SAS-6 expression. Ras-transformed MCF10AT1 cells (Miller, 1996) were kindly provided by the Barbara Ann Karmanos Cancer Institute (Detroit, MI, USA) and maintained in DMEM/F12 supplemented with 5% horse serum, 20 ng/ml EGF, 0.5 mg/ml hydrocortisone, 100 ng/ml cholera toxin, 10 mg/ml insulin, and 1% penicillin/streptomycin. HMECs-hTERT (Clontech) were cultured in mammary epithelial cell growth medium with the addition of MEGM BulletKit (LONZA). HEK293T cells used for lentiviral transduction were maintained in DMEM (Sigma-Aldrich) supplemented with 10% FBS and 1% penicillin/streptomycin.

Single-cell clones of RPE-1 cells, HMECs, and MCF10AT1 expressing tetracycline-inducible WT SAS-6 or SAS-6ND were obtained via lentiviral gene transduction with the pLVX-Tet-On Advanced inducible gene expression system (Clontech). Lentiviruses were produced by transfecting 293T cells with the pLVX constructs together with packaging and envelope vectors (Clontech) using the calcium phosphate precipitation method. Stable expressors were derived by selection with 5 µg/ml puromycin (Sigma-Aldrich). Doxycycline was purchased from Sigma-Aldrich, and used at 1 µg/ml. Verteporfin (Cayman Chemical) was used at 10 µg/ml.

## Western blots

Cells were lysed in ice-cold RIPA buffer (Sigma-Aldrich) supplemented with a cocktail of protease and phosphatase inhibitors (Thermo Fisher Scientific). Lysates were sonicated and cleared by centrifugation at 14,000$g$ for 15 min. Protein content was determined using the DC Bio-Rad Protein Assay (Bio-Rad) following the manufacturer's instructions. Protein samples were subsequently denatured at 95°C for 10 min, separated by SDS–PAGE on a 4–12% polyacrylamide gradient gel, and transferred to a nitrocellulose membrane. The membrane was then blocked with 5% nonfat dry milk in Tris-buffered saline and 0.05% Tween-20 (TBST) for 1 h before an overnight incubation at 4°C with the indicated antibodies. After this, membranes were washed 3X in TBST before and

**Figure 3. SAS-6 invasion phenotype is associated with the activation of the YAP/TAZ pathway.**
**(A)** Gene set enrichment analysis of microarray data. Normalized enrichment score, nominal $P$-value (NOM pval), and false discovery rate q-value (FDR qval) are shown for each indicated gene set. Data are representative of two independent experiments. **(B)** Enrichment plot for the dataset "TEAD DEPENDENT YAP TARGET GENES COMMON BETWEEN YAP AND TAZ" showing an enrichment score curve. Normalized enrichment score: 1.65291. **(C)** Heat map with the list of genes driving the change. Note the increase in YAP downstream targets CTGF and CYR61. **(D)** qRT-PCR showing the mRNA levels of YAP/TAZ target genes, CTGF ($t$ test, $P < 0.05$) (left) and CYR61 ($t$ test, $P < 0.05$) (right). Ct values were normalized to GAPDH, and expression was calculated as a fold change relative to controls. Data represent the mean ± SEM of three independent experiments performed in triplicate. **(E)** Western blot showing the expression of SAS-6ND with and without doxycycline (indicated). **(F)** Immunostaining for YAP in cells transduced with SAS-6ND, treated with vehicle control (top) or doxycycline (lower panels). Note the increase in nuclear YAP in cells overexpressing SAS-6ND. Scale bar, 10 µm. **(G)** Quantification of the nuclear/cytoplasmic ratio in control cells (SAS-6ND, -dox) and cells overexpressing SAS-6ND (+dox). $t$ test, $P < 0.001$. Note that SAS-6ND–overexpressing cells have increased nuclear YAP. **(H)** Tukey's boxplots showing luciferase activity of YAP reporter 8xGTIIC-luc (Firefly/Renilla) indicative of YAP/TAZ activation in control or cells induced with doxycycline to express SAS-6ND ($t$ test, $P < 0.05$). **(I)** Western blot showing YAP protein levels upon treatment with vehicle control or verteporfin (10 µg/ml) in control cells or doxycycline-induced cells expressing SAS-6ND (indicated). A quantification for three independent experiments is shown below. **(J)** Collagen invasion assay in control or doxycycline-induced cells expressing SAS-6ND upon treatment with verteporfin (10 µg/ml). N = 4 experiments.

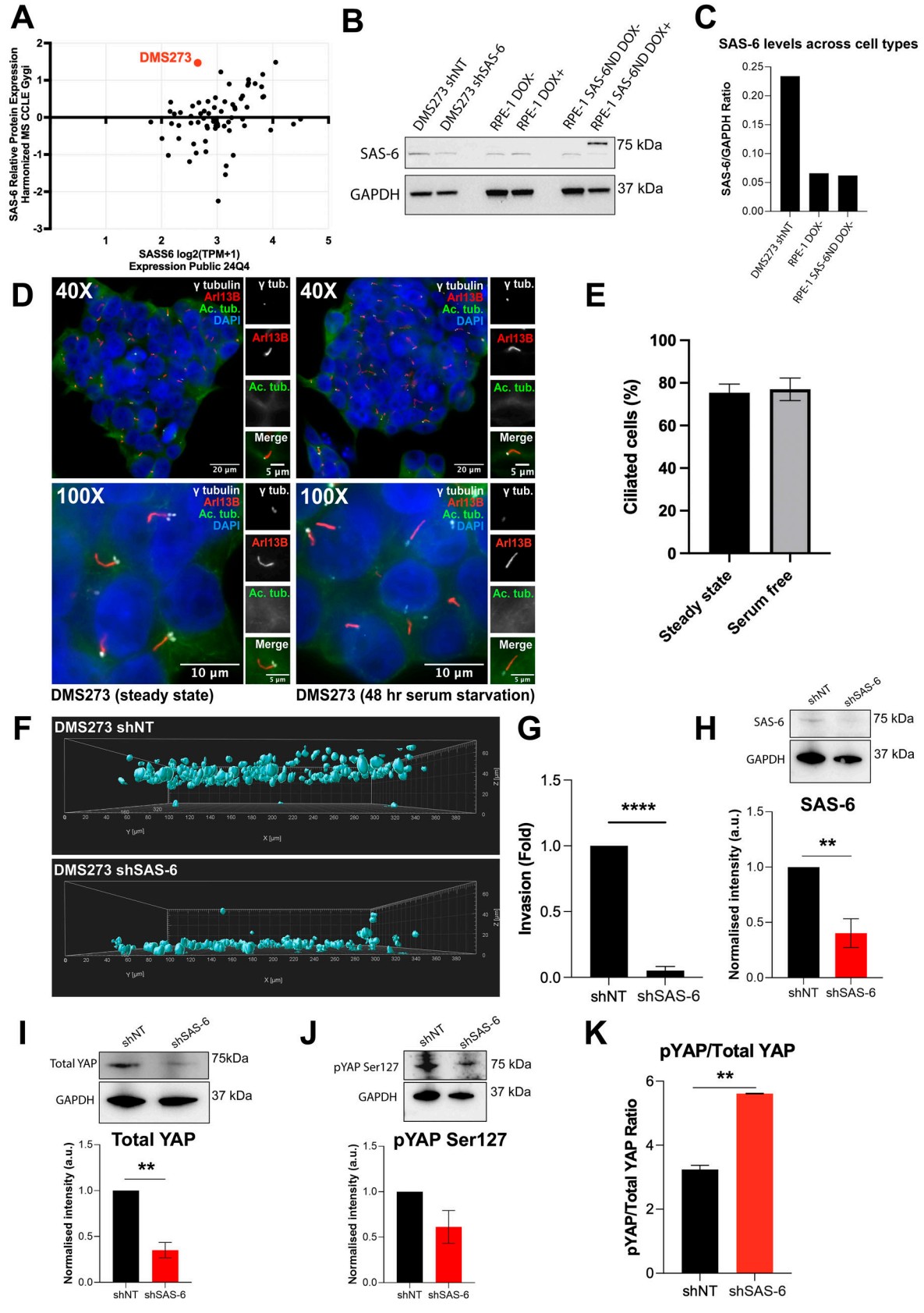

after the addition of HRP-conjugated anti-mouse or anti-rabbit IgG secondary antibodies (1:2,000; 7076; Cell Signaling Technology) for 1 h. Immunoreactivity was visualized using SuperSignal West Pico Chemiluminescent Substrate (34080; Thermo Fisher Scientific) and the Bio-Rad ChemiDoc XRS+ imaging system.

## Immunofluorescence

Cells grown on poly-L-lysine–coated coverslips were fixed in 4% PFA for 10 min at room temperature before permeabilization with 0.1% Triton X-100 in PBS and blocking with 3% (wt/vol) bovine serum albumin and 0.1% Triton X-100 in PBS for 5 min. Primary antibodies and Alexa Fluor 594 phalloidin (1:100; A12381; Molecular Probes) were diluted to desired concentrations in blocking solution and allowed to incubate for 1 h before three washes with 0.1% Triton X-100 in PBS (PBST). For centriolar SAS-6, PFA fixation was preceded by a 2 min permeabilization in PTEM buffer containing 20 mM PIPES (pH 6.8), 0.2% Triton X-100, 10 mM EGTA, and 1 mM $MgCl_2$. Goat secondary antibodies conjugated to Alexa Fluor 488, 594, or 680 (1:500 dilution; Thermo Fisher Scientific) were incubated for 1 h followed by three PBST washes and DAPI staining (Invitrogen) for DNA visualization. Coverslips were mounted with ProLong Gold Antifade Reagent (Invitrogen).

## Antibodies

The following antibodies were used: mouse monoclonal antibody to detect endogenous SAS-6 (1:500) was purchased from Santa Cruz Biotechnology (sc-81431). Overexpressed SAS-6 was detected with mouse anti-c-Myc (1:250; 9E10; 13-2500; Invitrogen) antibody, mouse anti-γ-tubulin (1:500; Santa Cruz Biotechnology, Inc.), and mouse anti-α-tubulin (1:2,000; Sigma-Aldrich). Cilia were detected with mouse anti-acetylated tubulin (1:2,000; Sigma-Aldrich) and with rabbit anti-Arl13B (1:500; 17711-1-AP; Proteintech). Centrioles were detected with mouse anti-centrin (1:500; 3E6; H00001070-M01; Abnova). Goat secondary antibodies conjugated to Alexa Fluor 488, 594, or 680 (1:500 dilution; Thermo Fisher Scientific) were used. For Western blot, polyclonal rabbit anti-GAPDH (EMD Millipore/AB2302) and monoclonal mouse anti-vinculin (1:10,000, V9131; Merck) antibodies were used. Total YAP was detected with YAP monoclonal antibody 63.7 (sc-101199) for both immunofluorescence (1:200) and Western blot (1:500). Phospho-YAP (S127) was detected with a rabbit PSer127 antibody (1:500, #4911; Cell Signaling Technology). SCLT1 was detected with a rabbit polyclonal antibody

(1:500; 14875-1-AP; Proteintech). Western blot secondary antibodies were HRP-conjugated rabbit or mouse anti-IgG antibodies (1:2,000; Cell Signaling).

## Image acquisition, processing, and analysis

Fluorescence images were acquired on an Axio Imager M2 microscope (Zeiss) equipped with 100x, 1.4 numerical aperture (NA) oil objective; an ORCA R2 camera (Hamamatsu Photonics); and ZenPro processing software (Carl Zeiss). Images were captured with similar exposure times and assembled into figures using Photoshop (CS5; Adobe).

Deconvolution microscopy was carried out with the DeltaVision Elite (Applied Precision) using an Olympus 100x, 1.4 NA oil objective; 405-, 488-, and 593-nm laser illumination; and standard excitation and emission filter sets. Raw images were acquired using a 0.2-$\mu$m z-step size and reconstructed in three dimensions with softWoRx 5.0.0 (Applied Precision) software. To determine the nuclear aspect ratio, height and length measurements of nuclei in the y–z plane were obtained using ImageJ.

Invasion assays were quantified in cells stained with phalloidin and DAPI and an ImageXpress Confocal High-Content Imaging System (Molecular Devices) and with a ZEISS Celldiscoverer 7 (CD7) microscope, using a Plan-Apochromat 20x/0.95NA air objective, with a 0.5x Optovar tube lens. Images were analyzed using Imaris 10.0 software.

Cell morphology analysis and transwell cell protrusion assays were imaged on a Zeiss LSM 710 confocal microscope with a 63x NA oil objective at optimal aperture settings. Four-times averaging per image was used. Image segmentation and cell morphology analysis (as shown in Fig 5) were performed using CellProfiler, ImageJ (OrientationJ), and MATLAB as described previously (Swiatlowska et al, 2022). Briefly, OrientationJ produces a weighted histogram for pixels per orientation. The weight is the coherency, which is defined through the ratio between difference and sum of the tensor eigenvalues and is bounded between 0 and 1, with 1 representing highly oriented structures (Rezakhaniha et al, 2012).

## Ciliogenesis experiments

Cells were plated in poly-L-lysine–coated coverslips in 3.5-cm plates at $0.4 \times 10^6$ cells per well and allowed to attach for 24 h. After this, cells were washed twice with serum-free medium, and left in serum-free medium for an additional 48 h. Cilia were detected by

---

**Figure 4. Cell invasion is sensitive to SAS-6 depletion in a lung cancer cell line.**
**(A)** Plot showing *SASS6* expression from RNA-seq data (DepMap Public 24Q4, log2[TPM+1]) versus SAS-6 relative protein expression (Harmonized MS CCLE Gygi), filtered by lung cancer. DMS273 is highlighted in red. **(B, C)** Western blot analysis showing SAS-6 expression across different cell types with densitometric quantification of SAS-6 relative to GAPDH for each cell line (shown in (C)). **(D)** Representative images of DMS273 cells grown in steady-state and serum-free conditions, taken at 40X and 100X. Cells are stained for cilium markers Arl13B (red) and acetylated tubulin (green). Centrioles are marked with γ-tubulin (white) and DNA with DAPI (blue). **(E)** Quantification of cilia (percent) in DMS273 cells grown in complete medium (steady state) or serum starved for 48 h (serum-free). Data are representative of three biological repeats. Error bars represent the SD. *t* test, $P > 0.05$. **(F)** Representative images from a collagen invasion assay in DMS273 cells transduced with nontargeting shRNA or SAS-6 shRNA (indicated). Cells were allowed to invade into a collagen matrix for 24 h, followed by fixation with PFA containing DAPI to stain nuclei. Images display rendered nuclear masks with their respective Z-positions and corresponding invasion fold quantification. Data are based on three biological replicates, each with three technical replicates. Images were analyzed using Imaris. **(G)** Quantification is shown in (G). **(H, I, J)** Representative Western blots and densitometric quantifications are shown for SAS-6 (H), total YAP (I), and phospho-YAP serine 127 (pYAP Ser127) (J). **(K)** Ratio of pYAP to total YAP for the experiments shown. All error bars represent the SD of the mean from three independent biological replicates. *t* test, $*P < 0.05$ or $**P < 0.01$ is indicated.

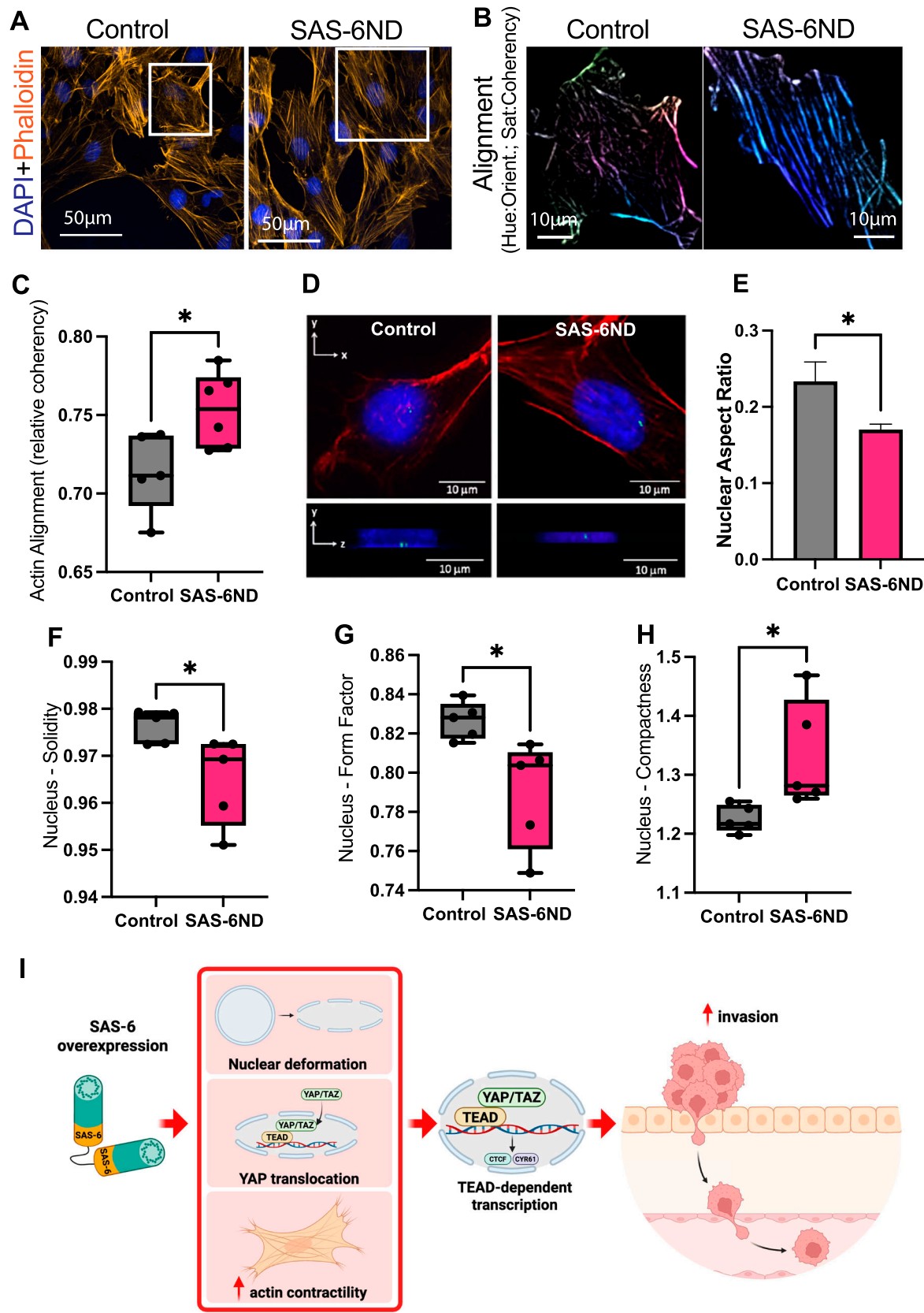

**Figure 5. SAS-6 promotes actin cytoskeleton changes and cell flattening.**
**(A)** Actin cytoskeleton staining in uninduced cells (control) or cells expressing SAS-6ND (indicated). Phalloidin-rhodamine is shown in orange. DNA is marked with DAPI (blue). **(A, B)** Actin alignment analysis showing cells from inset selection in (A). The hue is related to the orientation, and the saturation is coding for coherency. Note

staining with antibodies for acetylated tubulin and ciliary membrane protein Arl13B.

## Protrusion assays and 3D collagen invasion assays

Transwell protrusion formation assays were carried out on 3-$\mu$m pore transwell filters coated with 5 mg/ml collagen as described previously (Mardakheh et al, 2015). To quantify protrusions, a ratio of fluorescence intensity (measured with ImageJ) was calculated.

Collagen invasion assays were carried out as previously described (Sanz-Moreno et al, 2008). Briefly, cells were trypsinized and resuspended in serum-free liquid bovine type I collagen (3 mg/ml; 5005; CELLINK) in DMEM, dispensed into PerkinElmer black 96-well ViewPlates, and centrifuged at 188$g$ for 5 min to force cells toward the bottom of each well. Collagen polymerization proceeded for 3 h at 37°C, after which a final concentration of 10% FBS in DMEM/F-12 media was added to promote cell invasion into the collagen matrix. Each plate included vehicle-treated and doxycycline-treated cells as controls. Cells were fixed with a final concentration of 4% PFA for 24 h, permeabilized for 30 min with 0.5% Triton X-100 in PBS, and incubated with Alexa Fluor 594 phalloidin (1:100; A12381; Molecular Probes) and DAPI for 1 h. Confocal z-slices were captured every 10 $\mu$m (from 0 to 100 $\mu$m), and the number of nuclei in each plane was used to calculate an invasion index.

## Luciferase reporter assays

For the luciferase reporter assays, a TEAD reporter construct (8xGTIIC-luciferase, Plasmid #34615; Addgene) and a CMV-Renilla (pGL4.75[hRluc/CMV]; Promega) were used. Cells were seeded in six-well plates at 70% confluency. Cells were cotransfected with 2 $\mu$g of TEAD reporter and 100 ng of CMV-Renilla cDNA constructs using Lipofectamine 3000 according to the manufacturer's instructions. Cells were lysed 48 hours after transfection in 100 $\mu$l of lysis buffer (Promega). Aliquots of the cell lysates were used to read luciferase emission using Dual-Glo Luciferase Assay System (Promega) according to the manufacturer's instructions. Reporter firefly luciferase activity was normalized to Renilla activity.

## Small interfering RNA gene knockdown

Small interfering RNA (siRNA)–mediated gene knockdown was carried out using siGENOME SMARTPool siRNA (50 nM; Dharmacon) and Lipofectamine RNAiMAX Transfection Reagent (Invitrogen). Pooled sequences targeting SCLT1 were as follows: GAAAGAGACUGA GAGUUUA, GGUAGGAACUGACAUAUAU, GUAAAGAUCAGUACAAUGG, and GGCCCAGAUUCAUGUAUUU (M-016016-01-0005). Pooled sequences targeting YAP1 were as follows: GGUCAGAGAUACUUCUUAA, CCACCAAGCUAGAUAAAGA, GAACAUAGAAGGAGAGGAG, and GCACCU

AUCACUCUCGAGA (M-012200-00-0005). SCLT1 was down-regulated by performing two sequential transfections. Nontargeting siRNA was used as a control (50 nM; 1027281; QIAGEN). For SASS6 shRNA, we used a TRC shRNA (human) in a PLKO lentiviral vector, clone TRCN0000130315, sequence AAAGUCCAGUUGCUUAGUAGC (RHS3979-201849663; Horizon).

## RNA extraction

RNA was isolated from RPE-1 cells using the RNeasy mini kit (QIAGEN), and cDNA was generated using the High-Capacity cDNA Reverse Transcription Kit (Applied Biosystems) according to the manufacturer's protocols.

## GSEA

Gene expression was profiled using GeneChip Human Transcriptome Array 2.0. CEL file image data were converted to raw values using the R statistical language package "oligo" (Carvalho & Irizarry, 2010), available from https://www.r-project.org/. Pseudoimages were generated and inspected for artifacts. Data were normalized by robust multi-array average (RMA). For GSEA (Mootha et al, 2003; Subramanian et al, 2005), the data were transformed back to linear scale from log2 values. A custom gene set matrix curated to represent cell invasion was tested for enrichment using the online GSEA module version 17 of the GenePattern platform for reproducible bioinformatics (Reich et al, 2006). The specific settings applied in all analyses were as follows: number of permutations (1,000), permutation type (gene set), enrichment statistic (weighted), and metric for ranking genes ($t$ test). Tables show the normalized enrichment score, nominal $P$-value, and false discovery rate q-values. The list of the specific gene sets analyzed and their sources are available in Table S2.

## Reverse transcription and quantitative polymerase chain reaction (qRT-PCR)

Thermocycling was performed on the QuantStudio 7 Flex Real-Time PCR machine (Applied Biosystems) using TaqMan Universal PCR Master Mix (4366072; Applied Biosystems) and predesigned TaqMan probes (Applied Biosystems) for reference gene glyceraldehyde-3-phosphate dehydrogenase (GAPDH; Hs99999905) and YAP downstream targets cysteine-rich angiogenic inducer 61 (CYR61; Hs00155479) and connective tissue growth factor (CTGF; Hs00170014). Relative quantification was performed according to the ΔΔCT method (relative quantification, RQ = $2^{-\Delta\Delta CT}$), and expression levels of target genes were normalized to GAPDH.

---

more aligned pixels and higher coherency values in cells overexpressing SAS-6ND (right panel). **(C)** Actin alignment quantification with CellProfiler shows increased alignment in SAS-6ND–expressing cells. Data are representative of 6 independent experiments. **(D)** Three-dimensional rendering of cells overexpressing SAS-6ND. Cross section is shown where SAS-6ND–expressing cells show decreased nuclear height. **(D, E)** Quantification of the nuclear aspect ratio for cells shown in (D), as a ratio of the vertical to horizontal axis of the nucleus. Data are representative of three independent experiments. **(F, G, H)** CellProfiler quantifications show a significant decrease in nuclear solidity (as a ratio of the nuclear area/convex hull area) (F), and nuclear form factor (as a ratio of the area/perimeter) (G), and a concomitant increase in nuclear compactness (H), in SAS-6ND cells. $t$ test, $P < 0.05$ for all graphs is shown. Data are representative of six independent experiments. **(I)** Schema of SAS-6 effects on the actin cytoskeleton, YAP/TAZ pathway, and cell invasion.

### TCGA data analysis

Preprocessed TCGA mRNA abundance and clinical data were downloaded from TCGA DCC (GDAC), release: 2016_01_28. Patient groups were established by median dichotomizing *SASS6* expression profile resulting in low-expression and high-expression groups. The Cox proportional hazards model was used to estimate HR and 95% CIs with a *P*-value estimated using the log-rank test.

### Statistical analysis

Statistical tests were performed with GraphPad Prism. Statistical analyses, sample sizes, and error bars are defined in the figure legends. Paired *t* tests were performed to compare two experimental conditions. *P*-values equal to or less than 0.05 are indicated using asterisks: *$P \leq 0.05$, **$P \leq 0.01$, ***$P \leq 0.001$, ****$P \leq 0.0001$.

## Data Availability

Microarray data have been deposited in the EMBL European Bioinformatics Institute database repository (https://www.ebi.ac.uk/biostudies/) with the accession number E-MTAB-13783. Additional data generated or analyzed in this study will be made available upon request.

## Supplementary Information

## Acknowledgements

We thank Andrew Holland (Johns Hopkins School of Medicine) for discussions about this work. We thank Faraz Mardakheh (University of Oxford) for discussions and reagents, and Khine Nyein Myint (King's College London) and Elefterios Kostaras (Institute of Cancer Research) for reagents. BE Tanos is funded by a Kidney Research UK Senior Fellowship and a June Hancock Mesothelioma Research Fund. TCGA results reported here are based, in part, upon data generated by The Cancer Genome Atlas pilot project established by the NCI and NHGRI. Information about TCGA and the investigators and institutions who constitute TCGA research network can be found at http://cancergenome.nih.gov/. We thank Thomas Waring and the Centre for Cell Imaging (CCI), University of Liverpool, for their help with image acquisition on the CD7 (grant code: MR/X013502/1) and guidance for image analysis.

### Author Contributions

E Hargreaves, R Collinson, AD Jenks, A Staszewski: data curation, formal analysis, investigation, visualization, validation, methodology, and writing—review and editing.
A Tsalikis, R Bodoque, M Arias-Garcia, Y Abdi, A Al-Malki: formal analysis, validation, investigation, and methodology.

Y Yuan, R Natrajan, S Haider, T Iskratsch, F Calvo: formal analysis, data curation, validation, investigation, visualization, and methodology.
NJ Palaskas, W-J Wang, S Godinho, I Vivanco: resources, investigation, data curation, and methodology.
T Zech: formal analysis, resources, supervision, validation, methodology, and writing—review and editing.
BE Tanos: conceptualization, resources, data curation, formal analysis, supervision, funding acquisition, validation, investigation, visualization, methodology, project administration, and writing—original draft, review, and editing.

### Conflict of Interest Statement

The authors declare that they have no conflict of interest.

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
