## [Reviewer comments · Life Science Alliance]

Life Science Alliance

Dysregulated SASS6 expression promotes increased ciliogenesis and cell invasion phenotypes

Eleanor Hargreaves, Rebecca Collinson, Andrew Jenks, Adina Staszewski, Athanasios Tsalikis, Raquel Bodoque, Mar Arias Garcia, Yasmin Abdi, Abdulaziz Al-Malki, Yinyin Yuan, Rachael Natrajan, Syed Haider, Thomas Iskratsch, Won-Jing Wang, Susana Godinho, Nicolaos Palaskas, Fernando Calvo, Igor Vivanco, Tobias Zech, and Barbara Tanos

DOI: <https://doi.org/10.26508/lsa.202402820>

Corresponding author(s): Barbara Tanos, Brunel University of London

Review Timeline:	Submission Date:	2024-05-13
	Editorial Decision:	2024-06-27
	Revision Received:	2025-05-25
	Editorial Decision:	2025-06-25
	Revision Received:	2025-07-21
	Accepted:	2025-07-23

Scientific Editor: Tim Fessenden

Transaction Report:

June 27, 2024

Re: Life Science Alliance manuscript #LSA-2024-02820-T

Dr. Barbara E Tanos
The Institute of Cancer Research, London
College of Health, Medicine and Life Sciences
Division of Biosciences
Uxbridge, London UB8 3PH

Dear Dr. Tanos,

Thank you for submitting your manuscript entitled "Dysregulated SASS6 expression promotes increased ciliogenesis and cell invasion phenotypes" to Life Science Alliance. The manuscript was assessed by expert reviewers, whose comments are appended to this letter. We invite you to submit a revised manuscript addressing the Reviewer comments.

Thank you for this interesting contribution to Life Science Alliance. We are looking forward to receiving your revised manuscript.

Sincerely,

B. MANUSCRIPT ORGANIZATION AND FORMATTING:

Reviewer #1 (Comments to the Authors (Required)):

This is an interesting manuscript describing the role of overexpressed SAS-6 in cell invasion and cilia formation. SAS-6 is degraded in G1 by the APC-Cdh1. The authors utilize a Doxycycline-inducible construct of SAS6 that is resistant to degradation by the APC-Cdh1 and report effects on cell invasion and ciliogenesis. They further show that cilia are required for the effects on invasion along with the upregulation of the activity of the Hippo pathway. They also report actin-mediated cytoskeletal and cell shape changes that are consistent with the increased invasion seen by the upregulation of SAS-6. The authors conclude that upregulation of SAS-6 in solid tumors is essential for cell invasion via effects that are not related to its established role in centriole duplication. The individual effects of the overexpression of SAS-6 on all these cellular functions are interesting and can have a significant impact on our understanding of 1) how upregulated SAS-6 can mediate tumorigenesis and metastasis and 2) the role of enhanced ciliogenesis in these properties. The findings are also novel. However, there are also several points that can be addressed in a revised manuscript.

Main points

- 1) It is not clear how enhanced ciliogenesis, increased Hippo activity, and cell shape changes are linked to each other. Do they operate in parallel or in a linear fashion to affect cell invasion? More work is needed to connect these changes using cell invasion as the functional endpoint. In Fig.4I actin cytoskeletal changes are placed upstream of YAP/TAZ translocation, but there is no experimental evidence for that.
- 2) A degradation deficient construct is used throughout. It is not clear what the overall question is. Is it that the authors are aiming at addressing the role of APC-Cdh1-mediated degradation of SAS-6 in G1 in cell invasion or the role of the upregulation of SAS-6 in general? This is important because, in tumor cells, the APC-Cdh1-mediated degradation of SAS-6 is operational. It is understandable that massive overexpression can evade APC-Cdh1-mediated degradation, but this needs to be shown in a model tumor cell line. In this regard, would it be possible for the authors to use a tumor line in which the APC-Cdh1 is inactivated and revert a possible cell invasion phenotype by downregulating SAS-6?
- 3) Authors have correlated the upregulation of SAS-6 in cancer with poor prognosis. However, the cell line chosen does not represent a cancer type identified as such. MCF-10AT1 is used as a representative cell line for breast cancer, but they don't include an invasive breast cancer line where SAS-6 is expected to be upregulated. SAS-6 is upregulated in adrenocortical carcinoma, glioma, kidney, liver, and lung cancer. It is suggested that cell line from at least one of the reported cancers be used to show ciliary defects, YAP/TAZ activation, and cell invasion phenotype.
- 4) Are SAS-6 levels elevated in MCF-10AT1 cells compared to the non-Ras transformed MCF-10A control cells? This experiment should be conducted to confirm the differential expression of SAS-6 between these cell lines to support the point that it is over-expressed in cancer cells. MCF-10A can be considered a normal cell line, as it does not display an invasive phenotype in immunodeficient mice. Is endogenous or non-degradable SAS-6 localized at the centrosome during the G1 phase in the MCF-10AT1 cell line?
- 5) If, indeed, SAS-6 is overexpressed in MCF-10AT1, does its knockdown/normalization mitigate the invasive phenotype of these cells, if they display an invasive phenotype?
- 6) Please add details of the antibodies used for YAP detection. If possible, include the use of phospho-specific (p-YAP) versus non-phospho-YAP antibodies to demonstrate translocation to the nucleus. Alternatively, Western blots from nuclear versus cytoplasmic fractions should be performed to illustrate YAP activation and localization.
- 7) Please add (empty vector + doxycycline treatment) as a control for experiments in Figures 3F, H, and J (also in Figure 2, if not included). This is crucial to ensure that the observed effects are specific to changes in SAS-6 levels.
- 8) In addition to pharmacological means (verteporfin) used to suppress Hippo activity, genetic ablation of key components of the pathway would strengthen the conclusions.
- 9) Please add quantification for Fig3I replicates.
- 10) Doxycycline was used for 6 days. However, efficient induction can be achieved within 24 hours. Why didn't the authors use induction during serum starvation to test for the effect of ectopic expression in G0/G1 to obtain more direct evidence for the role of SAS-6DN in cilia length control? The same applies to cell invasion assays. They could compare and contrast effects from long- and short-term inductions on their cellular phenotypes.

Reviewer #2 (Comments to the Authors (Required)):

This manuscript reported that changes in SAS6 protein levels result in increased numbers of cilia. More interestingly, they observe changes in cell invasion, actin, and nuclear morphology. RNA transcriptomics suggest that changes in the YAP/TAX pathway may be responsible.

These are important observations. Previous experiments using PLK4 result in centriole overamplification. One important but not discussed result is association of the invasion phenotype and centriole amplification that can give rise to genome instability. It would be worth adding data about centriole amplification.

Lines 170-183: This is a key statement since many of the phenotypes have been observed when PLK4 is overexpressed and associated with centriole amplification. I think it is important to have more data and discussion.

The finding of changes in actin and the nuclear envelope are important. How different is this result from PLK4 overexpression?

The abstract discusses SAS-6ND but the results show similar results with overexpression of wild-type SAS6. Line 117: The authors show that over-expression of wild-type SAS6 also produces extra cilia. The invasion assay also shows the same phenotype with the wild-type and 6ND version. For the actin and nuclear sections, were these experiments performed with both wild-type and 6ND. Since these are different, do the authors think they have different effects. What is the difference in the amount of protein? Please comment.

Lines 194-195. "We found that among the 195 genes driving the enrichment of the YAP/TAZ gene set were CTGF and CYR61". How do the authors know that these genes are driving it? To make this statement they would need to make mutants or knockdown these genes. They have a correlation and not data that show the genes/proteins are driving it.

Line 210: Was the actin cytoskeleton examined after treatment with the inhibitor (verteporfin)? This would be great to include.

Line 236: The authors state that changes in the actin cytoskeleton led to nuclear deformation. Is this based on results from other groups? How strong is this evidence. Please clarify

Rewriting changes:

There is redundancy and mixing of background and results. Below are several suggestions for rewriting.

1. Lines 68-84: The final paragraph is part background and part summary of the results. I would turn it into 2 paragraphs. A sentence about metastasis and YAP/TAZ as background would help as well. Lines 124-129 would fit better in the intro rather than the results. I would move it to this proposed paragraph
2. I would suggest that the authors could use line 108 to start the new paragraph. We investigated whether SAS6 has additional functions besides centrosome duplication in S-phase. They then could continue line 76. The analysis of the TCGA is a new result but it was unclear in the original paragraph and seemed it could be background.
3. Line 217. The role of SAS6 in organizing actin is an important observations. The paragraph is redundant, and results are spread out. Concentrating the "background" info would help. Below is simple rearrangement with some deletions. Possible replacement of Lines 210-234.
Multiple solid cancers have shown an upregulation of YAP which correlates with increased invasion, malignancy and relapse (Piccolo et al., 2023). The YAP/TAZ pathway is activated by mechanical forces (Piccolo et al., 2023) and by cell flattening, which was proposed to promote the opening of the Nuclear Pore Complex (NPC) leading to the nuclear accumulation of YAP (Elosegui-Artola et al., 2017). We examined the morphology of the actin cytoskeleton in cells overexpressing SAS-6. SAS-6ND expressing cells showed increased actin alignment and condensed stress fibers (Fig. 4A, B, and C). Analysis of the vertical and horizontal axes in the nucleus revealed a decreased nuclear aspect ratio, thus confirming cell flattening (Fig. 4D, E). Further analysis showed decreased nuclear solidity and nuclear form factor and increased nuclear compactness (Fig. 4F, G, H).
4. Line 235-257. I suggest adding a title phrase: Model and Summary

Minor comments

1. ABSTRACT. "In cycling cells, SAS-6 undergoes APC/Cdh1 20-mediated targeted degradation 21 by the 26S proteasome at the end of mitosis". There are no results that go with this sentence. It should be in the introduction.
2. "Other functions of SAS6 that remain to be uncovered": I find this confusing. Here are functions known: bld12 cells, mutant for SAS6, possess basal bodies with 8-11 triplets mostly lack cilia, and have defects in the placement of the cleavage furrow. (Nakazawa et al. 2007), which may be a future unique to Chlamydomonas. In Tetrahymena, TtSAS6a and TtSAS6b have diverged such that TtSAS6a acts as a structural component of basal bodies, whereas TtSAS6b influences the location of new basal body assembly (Culver et al., 2009). I am not sure what is meant by features to be uncovered
3. Lines 46-52. Are these really needed. It seems that these lines could be deleted
4. Line 55. "dysregulated" Do the authors mean it is overexpressed? Dysregulation could be of many types.
5. Line 65 stating that STIL is a substrate is redundant with Line 53
6. Line 74. This needs a reference.
7. Line 96: I think the authors mean cycling cells not dividing cells. I think of dividing cells as ones in mitosis. Is SAS6 levels referring to protein? If so, please say protein.
8. Figure 1 need to explain that the red is the Ken box.
9. Line 98. KEN box domain results would not result

10. Line 101 replace where: with the KEN box replaced
11. Remove It was previously shown by the Tsou Laboratory that since it is referenced at the end of the sentence
12. Line 129.I would start this paragraph with this line about examining the TCGA.I would delete "any" from Line 130.
13. Line 138-140.I would delete these lines as they are redundant with the next few lines.
14. Line 131 states. "re-examined". Have the authors performed this analysis before? If so, please include a reference or consider rewording.
15. In Figure EV2, the cancers are shown in red and light blue. The reason for this was unclear.
16. Line 155.Media is plural. I am assuming that only one type of serum-free medium was used. Thus, the word medium should be used in both places
17. Line 156. Delete. --- a period of
18. Line 158. Delete (fold).it is unclear what it means in this sentence
19. Line 160. Delete For this experiment
20. Line 160. Insert hyphen. Serum-free medium
21. Line 161 Delete in an incubator. Unless you want to indicate addition of CO2
22. Line 162. Media to medium
23. Line 163.Does PFA need to be defined
24. Scale bar for panel C. in Figure 2 is needed
25. Line 240-1. Delete Previous work by G. Gupta uncovered

Reviewer #3 (Comments to the Authors (Required)):

1) In this study, Hargreaves et al. examine the function of the SAS-6 protein in regulating ciliogenesis. While SAS-6 is an important architectural component of centrioles and its role in centriole assembly is well-studied, its function in ciliogenesis is mechanistically less well understood. Therefore, the findings of this study are important. In their study, the authors use an inducible promoter to drive SAS-6 expression. The authors note that elevating SAS-6 levels causes an increase in both cilia number as well as length and results in increased invasion. Furthermore, the authors demonstrate that the invasion phenotype observed upon SAS-6 overexpression can be suppressed by the simultaneous depletion of a ciliary protein, SCLT-1; thereby suggesting that this is a cilia-dependent effect. The authors also report the up-regulation of genes belonging to the YAP/TAZ pathway in response to SAS-6 overexpression. Importantly, lowering YAP levels was found to be sufficient to suppress the invasion phenotype caused upon SAS-6 overexpression, suggesting that the invasion phenotype is YAP/TAZ dependent. The authors conclude their studies by demonstrating that SAS-6 overexpression causes remodeling of the actin cytoskeleton and cell flattening, which likely contribute to the invasion phenotype. This is a rigorous and well-thought out study. However, some concerns need to be addressed prior to the acceptance of this paper.

2) Major Comments:

- Since the authors correlate SAS-6 overexpression to an increased invasion and a poor prognosis of cancer patients, do patient-derived cancer cell lines exhibiting SAS-6 overexpression also show SAS-6 localization to the basal bodies during G1 or exhibit increased ciliogenesis like the present study shows? How do the expression levels of SAS-6 in this study compare with SAS-6 expression levels in patient-derived cancer cell lines?
- Figure 2D: SAS-6 overexpression is known to cause centrosome amplification (Leidel et al, 2005). The authors report that in their study, they observed minimal levels of centrosome amplification. However, could this also contribute to the increased invasion phenotype that they observe in their system? The authors attempt to address this issue by knocking down the ciliary protein SCLT-1, which is able to suppress the invasive phenotype of SAS-6 overexpression. However, it is unclear whether SCLT-1 depletion also affects centriole number in addition to inhibiting ciliogenesis. Further, is the cell cycle profile of SAS-6 overexpressing cells identical in the presence and in the absence of SCLT-1? It would be helpful if the authors could quantify ciliogenesis in SAS-6 overexpressing cells in the presence and in the absence of SCLT-1.

3) Minor comments:

- Line 24: "I" of inhibition should be capitalized.
- Please ensure that references are formatted correctly.
- Kindly use "cilium" for singular and "cilia" for plural.
- Figure 1: Please include a key in the legend specifying D = Distal; P=Proximal
- Please specify what kind of t-test was used to analyze the data.
- Line 535: The actual figure 2B panels are labeled as 0 microns and 40 microns, respectively. However, the legend states that the panels show confocal images at 5 microns and 40 microns. Kindly clarify which of them is correct.
- Line 536: The spelling of "Hoechst" is incorrect.
- Line 560: Please capitalize "Cyr61".
- Figure 4E: The spelling of "Nuclear" on the Y-axis is incorrect.
- There are two previously published studies in *C. elegans* pertaining to the role of SAS-6 in ciliogenesis. Please discuss the findings of these studies so that readers have the proper context to interpret the findings of this current study.

POINT BY POINT RESPONSE

RE: HARGREAVES et al POINT BY POINT RESPONSE

Dysregulated SASS6 expression promotes increased ciliogenesis and cell invasion phenotypes

Reviewer #1 (Comments to the Authors (Required)):

This is an interesting manuscript describing the role of overexpressed SAS-6 in cell invasion and cilia formation. SAS-6 is degraded in G1 by the APC-Cdh1. The authors utilize a Doxycycline-inducible construct of SAS6 that is resistant to degradation by the APC-Cdh1 and report effects on cell invasion and ciliogenesis. They further show that cilia are required for the effects on invasion along with the upregulation of the activity of the Hippo pathway. They also report actin-mediated cytoskeletal and cell shape changes that are consistent with the increased invasion seen by the upregulation of SAS-6. The authors conclude that upregulation of SAS-6 in solid tumors is essential for cell invasion via effects that are not related to its established role in centriole duplication. The individual effects of the overexpression of SAS-6 on all these cellular functions are interesting and can have a significant impact on our understanding of 1) how upregulated SAS-6 can mediate tumorigenesis and metastasis and 2) the role of enhanced ciliogenesis in these properties. The findings are also novel. However, there are also several points that can be addressed in a revised manuscript.

Main points

1) It is not clear how enhanced ciliogenesis, increased Hippo activity, and cell shape changes are linked to each other. Do they operate in parallel or in a linear fashion to affect cell invasion? More work is needed to connect these changes using cell invasion as the functional endpoint. In Fig.4I actin cytoskeletal changes are placed upstream of YAP/TAZ translocation, but there is no experimental evidence for that.

We thank the reviewer for the kind comments and insight. As shown in the original Figure 4, we observe significant changes in the nuclear aspect ratio and nuclear flattening, which would be consistent with a change in tension and increased actin contractility/invasion. The Roca-Cusachs lab has previously demonstrated that these mechanical changes can support YAP nuclear translocation and TEAD -dependent transcription (PMID: 29107331). So, the original model was based on this. To avoid overinterpretation, we have simplified the model in Figure 4 (now Figure 5), and removed some of the epistatic relationships from our previous model.

However, we have also generated additional data on the epistatic relationship between SAS-6 and YAP. Using a cell invasion readout (as suggested by the reviewer), we have carried out experiments in DMS 273, a cancer cell line that has endogenous high levels of SAS-6 (as suggested by the reviewer in point #2). DMS 273 shows an invasive

phenotype, that is reverted upon SAS-6 downregulation (new Figure 4). Notably, SAS-6 downregulation in DMS 273 also results in a decrease in YAP levels (new Figure 4) and an increased ratio of phospho-YAP to total YAP (new Figure 4), consistent with YAP cytoplasmic retention. These results suggest that SAS-6 is upstream of YAP/TAZ pathway activation.

In the original manuscript we used verteporfin to interrogate the role of YAP on SAS-6-driven invasion. To validate this finding using a genetic approach, we decided to also carry out invasion assays using a previously-validated YAP-targeting siRNA (PMID: 29093443). Knockdown of YAP reversed the invasion observed upon ND-SAS-6 overexpression and was consistent with a change in cell morphology, including nuclear height and nuclear sphericity (new supplementary Figure 9). A new supplementary Figure 9 has been added with these invasion results and quantification. This would indicate that downregulating YAP, reverts the cell morphology changes caused by SAS-6 overexpression. However, as mentioned above, the existing evidence indicates that this may not be a strictly linear process. So, for simplicity, we have modified the model indicating that SAS-6 results in nuclear deformation, YAP translocation to the nucleus, increased actin contractility and that these events result in TEAD-dependent transcription and invasion.

2) A degradation deficient construct is used throughout. It is not clear what the overall question is. Is it that the authors are aiming at addressing the role of APC-Cdh1-mediated degradation of SAS-6 in G1 in cell invasion or the role of the upregulation of SAS-6 in general? This is important because, in tumor cells, the APC-Cdh1-mediated degradation of SAS-6 is operational. It is understandable that massive overexpression can evade APC-Cdh1-mediated degradation, but this needs to be shown in a model tumor cell line. In this regard, would it be possible for the authors to use a tumor line in which the APC-Cdh1 is inactivated and revert a possible cell invasion phenotype by downregulating SAS-6?

We thank the reviewer for this comment. We feel that our new data addressing this point has significantly increased the quality of our manuscript. First, we want to clarify that the aim of this paper is to describe and characterise the role for SAS-6 in cell invasion, which has never been shown before. To identify a relevant tumour cell line with endogenous high levels of SAS-6, we searched the DepMap database (<https://depmap.org>) for cell lines where protein levels were significantly above those of other cell lines with similar mRNA expression levels. The lung cancer cell line DMS 273 meets these criteria (new Figure 4) and represents one of the cancer types where SAS-6 overexpression is associated with poor survival. Staining of this cell line for cilia markers, shows that DMS 273 had significant ciliation, 80% both in the presence of serum and following serum starvation. To lower the expression of SAS-6 in these cells, we used a lentiviral shRNA for SAS-6, which reduced protein expression by 50% (new Figure 4B,H). These cells were used to carry out invasion assays. Strikingly, downregulation of SAS-6 completely reverted their invasive capabilities (Figure 4D) and also led to the downregulation of YAP. These data show that downregulation of

endogenous SAS-6 in an invasive cancer cell line with high levels of SAS-6 can suppress invasion.

3) Authors have correlated the upregulation of SAS-6 in cancer with poor prognosis. However, the cell line chosen does not represent a cancer type identified as such. MCF-10AT1 is used as a representative cell line for breast cancer, but they don't include an invasive breast cancer line where SAS-6 is expected to be upregulated. SAS-6 is upregulated in adrenocortical carcinoma, glioma, kidney, liver, and lung cancer. It is suggested that cell line from at least one of the reported cancers be used to show ciliary defects, YAP/TAZ activation, and cell invasion phenotype.

We thank the reviewer for this comment. As stated above, using tools from DepMap, we identified a lung cancer cell line (DMS 273) with higher SAS-6 protein expression compared to other cell lines with similar SAS-6 mRNA levels. DMS 273 showed a high rate of ciliogenesis, even in the absence of serum, and a highly invasive phenotype, which was reverted upon shRNA-mediated downregulation of SAS-6 (new Figure 4). We examined YAP levels in these cells, which are indicative of YAP activation, as YAP promotes its own transcription, and we observed a decrease in the total YAP levels upon SAS-6 downregulation (Figure 4), supporting a role for SAS-6 in promoting the activation of the YAP/TAZ pathway. Furthermore, upon SAS-6 downregulation, these cells presented an increased ratio of P-YAP/Total YAP, which is a signal for cytoplasmic YAP retention. So, in summary, in DMS 273 cells we have shown:

- Increased SAS-6
- Increased ciliation
- Increased YAP
- SAS-6-dependent invasion
- YAP-levels dependent of SAS-6

4) Are SAS-6 levels elevated in MCF-10AT1 cells compared to the non-Ras transformed MCF-10A control cells? This experiment should be conducted to confirm the differential expression of SAS-6 between these cell lines to support the point that it is over-expressed in cancer cells. MCF-10A can be considered a normal cell line, as it does not display an invasive phenotype in immunodeficient mice. Is endogenous or non-degradable SAS-6 localized at the centrosome during the G1 phase in the MCF-10AT1 cell line?

We thank the reviewer for this comment and apologise if there was lack of clarity on this point. We chose to use MCF-10AT1 (a Ras transformed cell line) which has been previously characterised by our co-author and collaborator Dr. Rachel Natrajan as a model of pre-invasive breast carcinoma (PMID: 27512948). This model allowed us to interrogate the transition into an invasive phenotype through overexpression of SAS-6 using our doxycycline-inducible system. The results shown in Figure 2 are in MCF-10AT1 cells stably transduced with our inducible SAS-6 constructs where the effects of SAS-6 overexpression can be assessed through doxycycline treatment (Figure 2C). We

feel that an inducible system like this is ideal to study SAS-6-driven cell invasion. We have slightly modified the text to improve clarity.

As per the reviewer suggestion, we examined the localization of SAS-6ND and endogenous SAS-6 in MCF10-AT1 in serum-starved cells and we can observe that, upon dox-induction, SAS-6ND localizes to both centrioles. The localization of endogenous SAS-6 was variable but also present in both centrioles in serum starved cells. We have included this as a Figure within this revision.

[Figure removed by editorial staff per authors' request]

5) If, indeed, SAS-6 is overexpressed in MCF-10AT1, does its knockdown/normalization mitigate the invasive phenotype of these cells, if they display an invasive phenotype?

As stated in the previous point, this cell line has been previously characterised as preinvasive so it is not expected to have increased SAS-6 levels. But, to answer the reviewer's question, our data show that induced SAS-6 overexpression results in increased cell invasion (Figure 2C). A western blot with the corresponding levels for Myc-tagged SAS-6 is shown in Figure 2C (lower panels). The advantage of this system, is the regulatable nature of SAS-6 overexpression.

We have also obtained and characterized DMS 273, a patient-derived, highly invasive metastatic lung cancer cell line. This cell line is also highly ciliated. In this cell line,

downregulation of SAS-6 led to a significant decrease in invasion and to the downregulation of YAP (new Figure 4).

6) Please add details of the antibodies used for YAP detection. If possible, include the use of phospho-specific (p-YAP) versus non-phospho-YAP antibodies to demonstrate translocation to the nucleus. Alternatively, Western blots from nuclear versus cytoplasmic fractions should be performed to illustrate YAP activation and localization.

Thank you for this comment. The information for these antibodies has been added.

For DMS 273, we examined both total YAP and phospho-YAP following downregulation of SAS-6. This is shown in supplementary Figure 4. We can see that downregulation of SAS-6 results in a significant decrease in total YAP levels. This is reflective of a disruption of the YAP/TAZ pathway and TEAD-dependent transcription, as YAP itself is a target transcribed downstream of TEAD (22659496). When we analyze the ratio of P-YAP to total YAP, we observe an increase in this ratio. This is indicative of increased cytoplasmic YAP-localization as phosphorylation in YAP S-127 by LATS is known to promote cytoplasmic retention (PMID: 17974916).

7) Please add (empty vector + doxycycline treatment) as a control for experiments in Figures 3F, H, and J (also in Figure 2, if not included). This is crucial to ensure that the observed effects are specific to changes in SAS-6 levels.

There is an empty vector control for the initial phenotype in supplementary figure 1. In addition to this, the invasion phenotype was tested in different clones (each derived from a single cell) expressing SAS-6ND in order to rule out potential artifacts. We want to emphasize that because these inducible models were generated from single-cell clones, the comparison between induced and uninduced cells (i.e. +/- doxycycline) measures the effects of a single change (SAS-6 overexpression) in an otherwise genetically identical pair. We have also found that doxycycline treatment alone does not affect the YAP nuclear/cytoplasmic ratio (Figure S8).

As already mentioned, we have also carried out the reverse type of experiment using DMS 273 cells, a lung epithelial cell line that has high SAS-6 protein. In these cells, downregulation of SAS-6 decreased invasion and YAP levels, supporting a direct role for SAS-6 in this phenotype.

8) In addition to pharmacological means (verteporfin) used to suppress Hippo activity, genetic ablation of key components of the pathway would strengthen the conclusions.

We thank the reviewer for this comment. Following the suggestion, we have now assessed the effect of RNAi-mediated YAP knockdown and found a significant downregulation of the SAS-6 mediated invasion phenotype. This is shown in supplementary Figure 9.

9) Please add quantification for Fig3I replicates.

A graph with this has been added in Figure 3.

10) Doxycycline was used for 6 days. However, efficient induction can be achieved within 24 hours. Why didn't the authors use induction during serum starvation to test for the effect of ectopic expression in G0/G1 to obtain more direct evidence for the role of SAS-6DN in cilia length control? The same applies to cell invasion assays. They could compare and contrast effects from long- and short-term inductions on their cellular phenotypes.

We induced the cells for 6 days, to ensure that SAS-6 protein reached all its potential localizations in the cell in all the cells. As per the reviewer suggestion, we took a closer look in serum starved cells. We used both RPE-1 and RPE-1-SAS-6ND, We compared doxycycline induction at 3 days versus 6 days. A quantification of the percentage of cilia showed that the increase in cilia number was only observed after 6 days in the SAS-6ND dox-induced condition (Supplementary Figure 3). We also examined cilia length, and this showed an increase in only in SAS-6ND expressing cells, which was already noticeable after 3 days, and significant both at 3 days and 6 days.

Reviewer #2 (Comments to the Authors (Required)):

This manuscript reported that changes in SAS6 protein levels result in increased numbers of cilia. More interestingly, they observe changes in cell invasion, actin, and nuclear morphology. RNA transcriptomics suggest that changes in the YAP/TAX pathway may be responsible.

These are important observations. Previous experiments using PLK4 result in centriole overamplification. One important but not discussed result is association of the invasion phenotype and centriole amplification that can give rise to genome instability. It would be worth adding data about centriole amplification.

Lines 170-183: This is a key statement since many of the phenotypes have been observed when PLK4 is overexpressed and associated with centriole amplification. I think it is important to have more data and discussion.

We thank the reviewer for the kind comments. We have included an additional sentence in the discussion about PLK4 and centriole amplification.

Additionally, we have included new centriole quantification data. Upon SAS-6 overexpression, we do not observe centriole amplification which would be comparable to that seen with Plk4 overexpression. This is shown in supplementary Figure 2. We quantified centriole numbers in both RPE-1 cells and RPE-1SAS-6 ND expressing cells.

These results show that SAS-6 expression is not sufficient to promote a significant increase in centriole numbers compared to Plk4 overexpression.

The finding of changes in actin and the nuclear envelope are important. How different is this result from PLK4 overexpression?

We agree, the changes in cell morphology are very interesting. Plk4 overexpression has been shown to promote actin protrusions and changes in the actin cytoskeleton due to changes in centrosome amplification that affect Rac activation (PMID: 24739973). We do not observe centrosome amplification in our experimental system that could be comparable to that of Plk4. We hypothesize that the effect we see is the result of a signaling event occurring at centrioles, we plan to explore this in a follow up paper.

The abstract discusses SAS-6ND but the results show similar results with overexpression of wild-type SAS6. Line 117: The authors show that over-expression of wild-type SAS6 also produces extra cilia. The invasion assay also shows the same phenotype with the wild-type and 6ND version.

For the actin and nuclear sections, were these experiments performed with both wild-type and 6ND. Since these are different, do the authors think they have different effects.

We thank the reviewer for the comment and apologise if we did not make this point sufficiently clear. We believe that a sufficiently high level of WT-SAS-6 overexpression overwhelms the APC/Cdh1 degradation machinery allowing breakthrough SAS-6 protein to exist throughout the cell cycle. This has been previously described for other APC/Cdh1 substrates such as SAG (PMID: 32905768). Therefore, we expected that overexpression of WT-SAS6 or SAS6-ND could have similar effects. Figure 2C (lower panels) shows a western blot for overexpressed SAS-6 and how it compares WT with SAS-6ND. Our data on ciliation confirms our prediction. The actin cytoskeleton sections were carried out with ND-SAS-6 only.

What is the difference in the amount of protein? Please comment.

We have included a western blot with the levels of endogenous SAS-6 in RPE-1, RPE-SAS-6ND and DMS 273. For the doxycycline treated cells, these are isogenic systems where the protein is only expressed upon doxycycline induction. Endogenous SAS-6 levels in DMS 273 cells are comparable to the doxycycline-treated RPE-1 expressing SAS-6ND. Densitometry has been carried out for the levels of protein, and this is shown in the new Figure 4.

Lines 194-195. "We found that among the 195 genes driving the enrichment of the YAP/TAZ gene set were CTGF and CYR61". How do the authors know that these genes are driving it? To make this statement they would need to make mutants or knockdown these genes. They have a correlation and not data that show the genes/proteins are driving it.

The reviewer has a very good point, we have not downregulated these genes and examined the invasion phenotype. We have rephrased this statement and stated “ We found that among the genes most upregulated in YAP/TAZ gene set were Cyr61 and CTCF...”, to highlight this finding.

Line 210: Was the actin cytoskeleton examined after treatment with the inhibitor (verteporfin)? This would be great to include.

We thank the reviewer for this observation. We tested cell compactness and cell form factor/solidity, which indicate changes in the shape of the cell. Cells became more compact and highly asymmetric (decreased solidity/form factor) upon doxycycline treatment.

It was rightly pointed out by reviewer #1 that verteporfin is a pharmacological inhibitor and that a genetic experiment might firm up our results. Therefore, rather than simply using this inhibitor we carried out an experiment using a YAP-targeting siRNA and assessed the changes in cell morphology. We observed that the changes in cell morphology caused by SAS-6 induction were suppressed by siRNA-mediated YAP knockdown, supporting a role for YAP in mediating this phenotype (Supplementary Figure 9).

Line 236: The authors state that changes in the actin cytoskeleton led to nuclear deformation. Is this based on results from other groups? How strong is this evidence. Please clarify

Yes, this is based on work by the lab of Pere Roca-Cusachs where the authors showed that nuclear flattening facilitates YAP translocation to the nucleus (PMID: 29107331, PMID: 39120491). The NPC itself has been shown to respond to cell tension by changing diameter in response to stress (PMID: 34762489). This has been clarified in the text.

Rewriting changes:

There is redundancy and mixing of background and results. Below are several suggestions for rewriting.

1. Lines 68-84: The final paragraph is part background and part summary of the results. I would turn it into 2 paragraphs. A sentence about metastasis and YAP/TAZ as background would help as well. Lines 124-129 would fit better in the intro rather than the results. I would move it to this proposed paragraph

2. I would suggest that the authors could use line 108 to start the new paragraph. We investigated whether SAS6 has additional functions besides centrosome duplication in S-phase. They then could continue line 76. The analysis of the TCGA is a new result but it was unclear in the original paragraph and seemed it could be background.

3. Line 217. The role of SAS6 in organizing actin is an important observations. The paragraph is redundant, and results are spread out. Concentrating the "background"

info would help. Below is simple rearrangement with some deletions. Possible replacement of Lines 210-234.

Multiple solid cancers have shown an upregulation of YAP which correlates with increased invasion, malignancy and relapse (Piccolo et al., 2023). The YAP/TAZ pathway is activated by mechanical forces (Piccolo et al., 2023) and by cell flattening, which was proposed to promote the opening of the Nuclear Pore Complex (NPC) leading to the nuclear accumulation of YAP (Elosegui-Artola et al., 2017). We examined the morphology of the actin cytoskeleton in cells overexpressing SAS-6. SAS-6ND expressing cells showed increased actin alignment and condensed stress fibers (Fig. 4A, B, and C). Analysis of the vertical and horizontal axes in the nucleus revealed a decreased nuclear aspect ratio, thus confirming cell flattening (Fig. 4D, E). Further analysis showed decreased nuclear solidity and nuclear form factor and increased nuclear compactness (Fig. 4F, G, H).

4. Line 235-257. I suggest adding a title phrase: Model and Summary

We thank the reviewer for these insightful comments. These writing suggestions have been incorporated.

Minor comments

1. ABSTRACT. "In cycling cells, SAS-6 undergoes APC^{Cdh1} 20-mediated targeted degradation 21 by the 26S proteasome at the end of mitosis". There are no results that go with this sentence. It should be in the introduction.

We believe this sentence is necessary, as it briefly explains the rationale for the experiments shown. We have now modified the abstract so that it is more connected to the description of the results.

2. "Other functions of SAS6 that remain to be uncovered": I find this confusing. Here are functions known: bld12 cells, mutant for SAS6, possess basal bodies with 8-11 triplets mostly lack cilia, and have defects in the placement of the cleavage furrow. (Nakazawa et al. 2007), which may be a future unique to Chlamydomonas. In Tetrahymena, TtSAS6a and TtSAS6b have diverged such that TtSAS6a acts as a structural component of basal bodies, whereas TtSAS6b influences the location of new basal body assembly (Culver et al., 2009). I am not sure what is meant by features to be uncovered

We apologize for the confusion, this was referring at additional functions in addition to centriolar function. The text has been modified accordingly.

3. Lines 46-52. Are these really needed. It seems that these lines could be deleted These lines have been removed.

4. Line 55. "disregulated" Do the authors mean it is overexpressed? Dysregulation could be of many types.

This has been corrected with "overexpression".

5. Line 65 stating that STIL is a substrate is redundant with Line 53 This has been modified.

6. Line 74. This needs a reference. A reference has been added

7. Line 96: *I think the authors mean cycling cells not dividing cells. I think of dividing cells as ones in mitosis.*

Thank you, this has been corrected

Is SAS6 levels referring to protein? If so, please say protein. This has been added.

8. *Figure 1 need to explain that the red is the Ken box. This has been explained in the legend*

9. *Line 98. KEN box domain results would not result This change has been made*

10. *Line 101 replace where: with the KEN box replaced This change has been made*

11. *Remove It was previously shown by the Tsou Laboratory that since it is referenced at the end of the sentence This change has been made*

12. *Line 129. I would start this paragraph with this line about examining the TCGA. I would delete "any" from Line 130. This has been corrected.*

13. *Line 138-140. I would delete these lines as they are redundant with the next few lines. This paragraph has been corrected.*

14. *Line 131 states. "re-examined". Have the authors performed this analysis before? If so, please include a reference or consider rewording. This has been corrected.*

15. *In Figure EV2, the cancers are shown in red and light blue. The reason for this was unclear. This has been corrected to only black.*

16. *Line 155. Media is plural. I am assuming that only one type of serum-free medium was used. Thus, the word medium should be used in both places*

17. *Line 156. Delete. --- a perod of Done*

18. *Line 158. Delete (fold). it is unclear what it means in this sentence Done*

19. *Line 160. Delete For this experiment Done*

20. *Line 160. Insert hyphen. Serum-free medium Done*

21. *Line 161 Delete in an incubator. Unless you want to indicate addition of CO2 Done*

22. *Line 162. Media to medium Done*

23. *Line 163. Does PFA need to be defined This has been defined.*

24. *Scale bar for panel C. in Figure 2 is needed Done*

25. *Line 240-1. Delete Previous work by G. Gupta uncovered Done*

We thank the reviewer for these insightful points, these minor corrections have been addressed.

Reviewer #3 (Comments to the Authors (Required)):

1) In this study, Hargreaves et al. examine the function of the SAS-6 protein in regulating ciliogenesis. While SAS-6 is an important architectural component of centrioles and its role in centriole assembly is well-studied, its function in ciliogenesis is mechanistically less well understood. Therefore, the findings of this study are important. In their study, the authors use an inducible promoter to drive SAS-6 expression. The authors note that elevating SAS-6 levels causes an increase in both cilia number as well as length and results in increased invasion. Furthermore, the authors demonstrate that the invasion phenotype observed upon SAS-6 overexpression can be suppressed by the simultaneous depletion of a ciliary protein, SCLT-1; thereby suggesting that this is a cilia-dependent effect. The authors also report the up-regulation of genes belonging to the YAP/TAZ pathway in response to SAS-6 overexpression. Importantly, lowering YAP levels was found to be sufficient to suppress the invasion phenotype caused upon SAS-6

overexpression, suggesting that the invasion phenotype is YAP/TAZ dependent. The authors conclude their studies by demonstrating that SAS-6 overexpression causes remodeling of the actin cytoskeleton and cell flattening, which likely contribute to the invasion phenotype. This is a rigorous and well-thought out study. However, some concerns need to be addressed prior to the acceptance of this paper.

2) Major Comments:

- Since the authors correlate SAS-6 overexpression to an increased invasion and a poor prognosis of cancer patients, do patient-derived cancer cell lines exhibiting SAS-6 overexpression also show SAS-6 localization to the basal bodies during G1 or exhibit increased ciliogenesis like the present study shows? How do the expression levels of SAS-6 in this study compare with SAS-6 expression levels in patient-derived cancer cell lines?

We thank the reviewer for these kind words and for the encouragement mentioning that: "This is a rigorous and well-thought-out study".

Following on the reviewer's suggestion, we obtained a patient-derived cell line: DMS 273. DMS273 is a cell line originally derived from a female lung cancer patient (PMID: 2986244). We found that DMS 273 cells had high SAS-6 protein expression when compared to endogenous levels in RPE-1 cells (new Figure 4). We examined ciliogenesis in DMS 273 cells, and found that it is indeed a highly ciliated cell line, even in the presence of serum. We also now show that downregulating SAS-6 in this cell line impairs the invasion phenotype. (New Figure 4).

- Figure 2D: SAS-6 overexpression is known to cause centrosome amplification (Leidel et al, 2005). The authors report that in their study, they observed minimal levels of centrosome amplification. However, could this also contribute to the increased invasion phenotype that they observe in their system?

We agree with the reviewer that centrosome amplification has been shown to promote invasion. This was shown by Professor Susana Godinho, who is also an author in this paper. In our case, SAS-6 was only overexpressed for 6 days, to allow efficient incorporation to centrioles. This did not result in centrosome amplification, as now shown in Supplementary Figure 2 and Supplementary Figure 6. A statement highlighting this result has been added in the text, also clarifying that, given these results, the invasion phenotype is unlikely the result of centrosome amplification.

The authors attempt to address this issue by knocking down the ciliary protein SCLT-1, which is able to suppress the invasive phenotype of SAS-6 overexpression. However, it is unclear whether SCLT-1 depletion also affects centriole number in addition to inhibiting ciliogenesis.

Thank you for the suggestion, we did not observed changes in centriolar numbers in the absence of SCLT1. This is shown both looking at centrin foci, as well as CP110 foci.

(Supplementary Figure 6). This figure includes a western blot showing protein levels and a quantification of cilia and cilia length. For CP110, it can be observed that cells presenting a cilium only have 1 foci, but that downregulation of SCLT1 shows two foci, as the lack of SCLT1 has been shown to impair CP110 removal.

Further, is the cell cycle profile of SAS-6 overexpressing cells identical in the presence and in the absence of SCLT-1?

As per the reviewer suggestion, we analyzed the cell cycle profile in cells where SCLT1 had been downregulated. This showed that removing SCLT1 did not change the cell cycle profile. This is shown in Supplementary Figure 5.

It would be helpful if the authors could quantify ciliogenesis in SAS-6 overexpressing cells in the presence and in the absence of SCLT-1.

This has been quantified and added to Supplementary Figure 6.

3) Minor comments:

- *Line 24: "I" of inhibition should be capitalized.* This has been corrected.
- *Please ensure that references are formatted correctly.* This has been corrected.
- *Kindly use "cilium" for singular and "cilia" for plural.* This has been corrected.
- *Figure 1: Please include a key in the legend specifying D = Distal; P=Proximal* This has been added.

- *Please specify what kind of t-test was used to analyze the data.*

This has been clarified in the methods section.

- *Line 535: The actual figure 2B panels are labeled as 0 microns and 40 microns, respectively. However, the legend states that the panels show confocal images at 5 microns and 40 microns. Kindly clarify which of them is correct.* This has been clarified.
- *Line 536: The spelling of "Hoechst" is incorrect.* Done.

- *Line 560: Please capitalize "Cyr61".* Done

- *Figure 4E: The spelling of "Nuclear" on the Y-axis is incorrect.*

This has been modified.

- *There are two previously published studies in C. elegans pertaining to the role of SAS-6 in ciliogenesis. Please discuss the findings of these studies so that readers have the proper context to interpret the findings of this current study.*

The findings on C. elegans and ciliogenesis have been incorporated to the discussion.

June 25, 2025

RE: Life Science Alliance Manuscript #LSA-2024-02820-TR

Dr. Barbara E Tanos
Brunel University of London
College of Health, Medicine and Life Sciences
Department of Biosciences
Uxbridge, London UB8 3PH
United Kingdom

Dear Dr. Tanos,

Thank you for submitting your revised manuscript entitled "Dysregulated SASS6 expression promotes increased ciliogenesis and cell invasion phenotypes". As you will see, reviewers are now satisfied and recommend publication. We would be happy to publish your paper in Life Science Alliance pending final revisions necessary to meet our formatting guidelines.

- Please be sure that the authorship listing and order is correct.
- Please add a Category for your manuscript in our system.
- Please add the X and Bluesky handles of your host institute/organization, as well as your own and/or one of the authors, in our system.
- Please add an Author Contributions section to your main manuscript text.
- Please add a Conflict of Interest statement to your main manuscript text.
- Please add a Data Availability statement to your main manuscript text.
- Please add your main, supplementary figure, and table legends to the main manuscript text after the references section.
- Please ensure the manuscript sections are aligned with LSA's formatting guidelines. For example, please separate the Figure legends and Supplemental Figure legends into separate sections. Please also separate the Results and Discussion into separate sections.
- Please remove label A from the figure S5 and its legend, since this figure has only one panel.
- Please use the [10 author names, et al.] format in your references (i.e., limit the author names to the first 10).
- There is a call-out for figure 5J, which doesn't have this panel; please correct.
- Please add callouts for Figures S2A-C; S3A-C; S6A-G; 3E; S7C; S8A-D; S9A-I and 4A, G-K to your main manuscript text.
- Please add molecular weight markers to the blots in figures 2, S6, 3, S9, and 4.

A. FINAL FILES:

-- Summary blurb (enter in submission system): A short text summarizing in a single sentence the study (max. 200 characters including spaces). This text is used in conjunction with the titles of papers, hence should be informative and complementary to

the title. It should describe the context and significance of the findings for a general readership; it should be written in the present tense and refer to the work in the third person. Author names should not be mentioned.

B. MANUSCRIPT ORGANIZATION AND FORMATTING:

Sincerely,

Reviewer #1 (Comments to the Authors (Required)):

The authors have been responsive to our previous comments and have addressed the major points raised. The inclusion of the DMS273 line and data showing the dependance of the invasion properties of this line on SAS-6, is a major improvement. We recommend the manuscript for publication in LSA.

Reviewer #2 (Comments to the Authors (Required)):

This is an interesting paper about overexpression of SAS6 amd its effects on cilia number, cell shpape, and invasion. The authors have addressed all of my concerns and it has greatly improved the manuscript.

July 23, 2025

RE: Life Science Alliance Manuscript #LSA-2024-02820-TRR

Dr. Barbara E Tanos
Brunel University of London
College of Health, Medicine and Life Sciences
Department of Biosciences
Uxbridge, London UB8 3PH
United Kingdom

Dear Dr. Tanos,

Thank you for submitting your Research Article entitled "Dysregulated SASS6 expression promotes increased ciliogenesis and cell invasion phenotypes". It is a pleasure to let you know that your manuscript is now accepted for publication in Life Science Alliance. Congratulations on this interesting work.

DISTRIBUTION OF MATERIALS:

Again, congratulations on a very nice paper. I hope you found the review process to be constructive and are pleased with how the manuscript was handled editorially. We look forward to future exciting submissions from your lab.

Sincerely,
